# GHz acousto-optic angular momentum with tunable topological charge

A. Pitanti[1,2,3] ✉, N. Ashurbekov[1], I. dePedro-Embid[1], M. Msall[1,4] & P. V. Santos [1] ✉

Controlling the symmetry of optical and mechanical waves is pivotal to their full exploitation in technological applications and topology-linked fundamental physics experiments. Leveraging on the control of orbital angular momentum, we introduce here a device forming acoustic vortices which can impart an orbital angular momentum modulation at super-high-frequency on reflected light beams. Originated by shape-engineering of a single-contact bulk acoustic wave resonator, acoustic vortices are generated in a wide band of frequencies around 4 GHz with topological charge ranging from 1 to beyond 13 tunable by the device geometry and/or excitation frequency. With all electrical control and on-chip integration our device offers compact solutions for angular-momentum-based light communication, three-dimensional particle manipulation, as well as alternative interaction schemes for optomechanical devices.

The observation that helical phase fronts of light carry an orbital angular momentum[1] ignited a wide research activity focused on harnessing the full vectorial properties of photons, adding spin and orbital angular momentum (SAM/OAM) to the scalar amplitude and phase[2,3]. The benefits of angular momentum control include new, high-bandwidth modulation schemes that exploit the OAM basis in optical communications[4], the possibility of trapping and rotational manipulation of particles via optical tweezers[5], higher dimensionality interferometric sensing[6], all of these on top of several fundamental physical effects linked to topology[7,8] and other applications[3].

The impressive advancements of vectorial and chiral photonics have been recently mirrored by fascinating results in the investigation of acoustic and elastic waves, which have evolved from the main component of radio-frequency filters[9] and oscillators[10] to tools for the manipulation of single excitations in quantum information applications[11]. The route for achieving a full vectorial control of elastic waves has been mostly focused on a frequency range between tens of Hz and a few MHz[12]: here, the angular momentum-carrying waves can propagate in fluids and have been predominantly used to exert forces on small particles, which can be trapped in three-dimensional potentials and/or subjected to a torque for rotational actuation[13–15].

Coupling together vectorial light and vibrations at ultra-high (0.3–3 GHz - UHF) and super-high (3–30 GHz - SHF) frequencies reveals a new class of on-chip, chiral and angular-momentum based acousto-optic devices, which can unveil several intriguing effects, lead by angular momentum-based modulation of light. These add to the already demonstrated surface acoustic wave amplitude/phase modulators[16,17], enriching the schemes for acousto-optic light manipulation, which participate in a general trend towards exploiting ultra-degree-of-freedom structured light for ultracapacity information carriers[18]. These have already showed promising results in OAM-based multiplexing[19] and encryption[20] as well as increased channel transfer capacity with all-optical vortex[21–23] and more complex topological structures[24,25]. Conversely, UHF and SHF chiral and helical acoustics provide innovative methods for the topological control of mechanical waves, expanding on concepts of wave manipulation through acoustic metasurfaces[26,27] and offering powerful tools for the control of hybrid quantum systems as, for exampnle, acoustic exciton polaritons[28].

In this manuscript we introduce a versatile approach for generalized OAM acousto-optical interactions by manipulating the shape of a piezoelectric bulk acoustic wave resonator (BAWR) to induce the generation of elastic vortex beams within a solid-state substrate. The vortices are generated in a wide frequency band in the GHz range and used to modulate the angular momentum of reflected light, introducing a new approach for the generation of OAM-carrying optical beams. The vortex topological invariant, also called topological charge ($\ell$), of the acousto-optical vortex beams can be externally controlled in a very wide range in a single device (from $\ell = 1$ to 13 and beyond) by

[1]Paul-Drude-Institut für Festkörperelektronik, Leibniz-Institut im Forschungsverbund Berlin e. V., Berlin, Germany. [2]University of Pisa, Dipartimento di Fisica E. Fermi, Pisa, Italy. [3]CNR-Istituto Nanoscienze, Pisa, Italy. [4]Department of Physics and Astronomy, Bowdoin College, Brunswick, ME, USA. ✉e-mail: alessandro.pitanti@unipi.it; santos@pdi-berlin.de

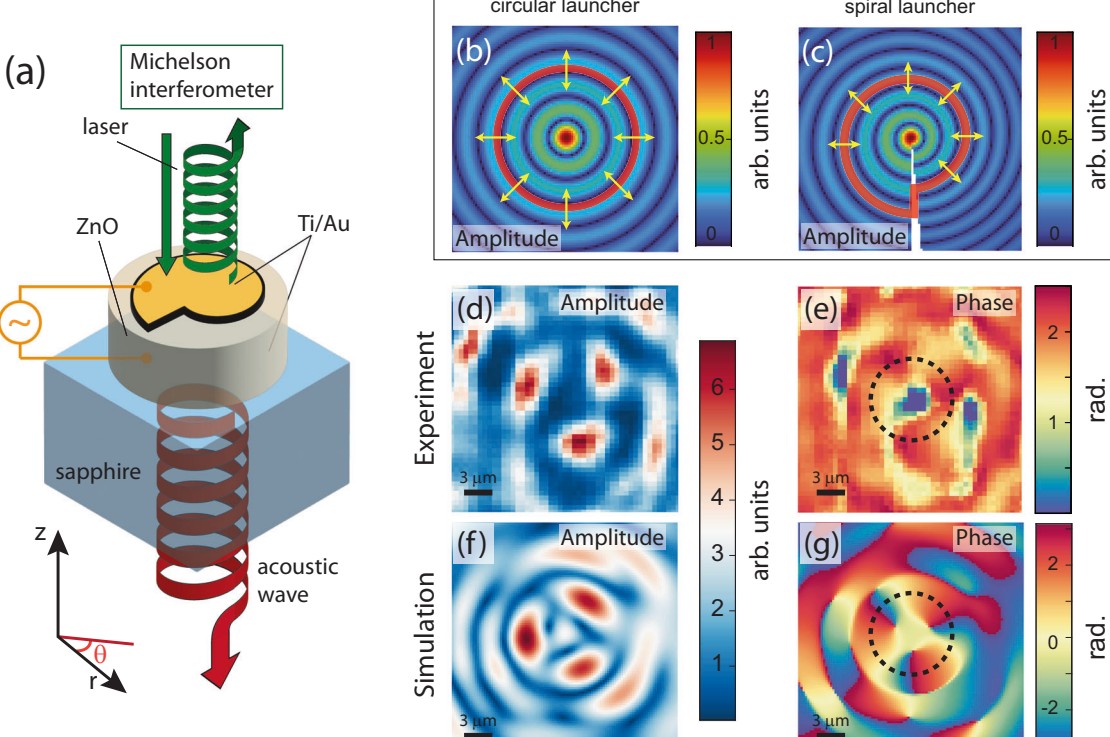

**Fig. 1 | Acousto-optical generation of chiral optical beams. a** Device concept. A properly engineered bulk acoustic wave resonator (BAWR) launches an acoustic vortex into the substrate and modulates in time the OAM of a reflected optical beam. The BAWR is engineered starting from a circular contact (with the expected acoustic field sketched in (**b**)) that is perturbed into a spiral (expected field sketched in (**c**)). **d**, **e** Experimental amplitude and phase map of a vortex with $\ell = 3$ at 1.139 GHz. **f**, **g** Simulated maps for the same drive and device geometry extracted from full, 3D simulations.

simply changing the driving frequency, with the additional option of operating at negative or within an increased $\ell$ range by introducing simple geometrical modifications to the basic device structure. The vertical profile of the generated acoustic field is qualitatively similar to a reflective spiral phase plate[29], rotating upon its axis at the mechanical frequency; its generalized interaction with an illuminating wavefront, more specifically based on a moving-boundary effect, reflects a wave with a RF rotational angular momentum[30], realizing a system similar to the ones employing macroscopic spinning optical elements for the detection of rotational Doppler's effect[31], albeit at super-high rotational frequencies. The $\ell$ tunability, broadband light interaction and high operating frequency (also resulting in a reduced footprint) establish our platform as a new class of versatile acousto-optic devices for hybrid systems, advanced acoustics and light-based microscopy and telecommunications, with distinct characteristics with respect to dynamics acousto-optics manipulation in optical fibers[32] or liquid crystals[33]. The possibility of superimposing acoustic modes with different $\ell$'s via frequency-based multiplexing opens the route to time-modulated, complex structured beams, which have been theoretically proposed for time modulated, spatially varying metasurfaces[34,35] and further advances the current technologies for optomechanical manipulation of objects and time-division multiple access in optical and quantum communication systems[35].

## Results and discussion

### Device concept

The basic device concept is displayed in Fig. 1a. A bulk acoustic wave resonator (BAWR) composed by a ZnO piezoelectric layer sandwiched between metallic contacts (see Methods for more details) is placed upon a double-polished sapphire substrate. A proper shaping of the BAWR contact is used to generate an acoustic vortex propagating across the substrate; shining light at the device top surface results in the creation of a time-varying OAM reflected optical beam. In our experiment the focused laser light is analyzed by a Michelson interferometer, which allows extraction of coherent field maps in the acousto-optical interaction plane.

In order to generate vortices, the BAWR contact shape must be modified to produce opportune phase profiles in the $\hat{r} - \hat{\theta}$ plane, see the reference axes in Fig. 1a, where the origin of coordinate system is placed in the center of the BAWR; a standard circular launcher, in fact, creates a radial wavefront which can be described, in a first approximation, by Bessel's functions of the first kind, $J_l(2\pi r/\lambda_a)$, where $\lambda_a$ is the radial acoustic wavelength, (see the top-view sketch of Fig. 1b). Considering the rigorous definition of $\ell$ as[3]:

$$\ell = \frac{1}{2\pi} \oint_{\gamma_c} \nabla \phi(\mathbf{r}) \mathbf{dr}, \tag{1}$$

where $\gamma_c$ is a tiny loop surrounding the singularity, it follows that the phase $\phi$ of a standard vortex field with topological charge $\ell$ linearly increases from 0 to $2\pi\ell$ along the path $\gamma_c$; considering an isotropic sound velocity distribution in the $\hat{r} - \hat{\theta}$ plane, the required phase profile can be obtained by modifying the circular launcher into the shape of an Archimedean Spiral (AS), as can be intuited by looking at the sketch of Fig. 1c, where the circular Bessel's function was deformed by adding a linear phase in the azimuthal direction. The AS is conveniently defined considering the trajectory of a point whose distance from the center linearly increases with the azimuthal angle $\hat{\theta}$. It has been previously employed to generate vortices in different systems including optical beams[36], hyperbolic polaritons[37], plasmons[38] and low-frequency sound waves[39]. For practical reasons, including broadband operation, we introduce the AS in the BAWR contact by

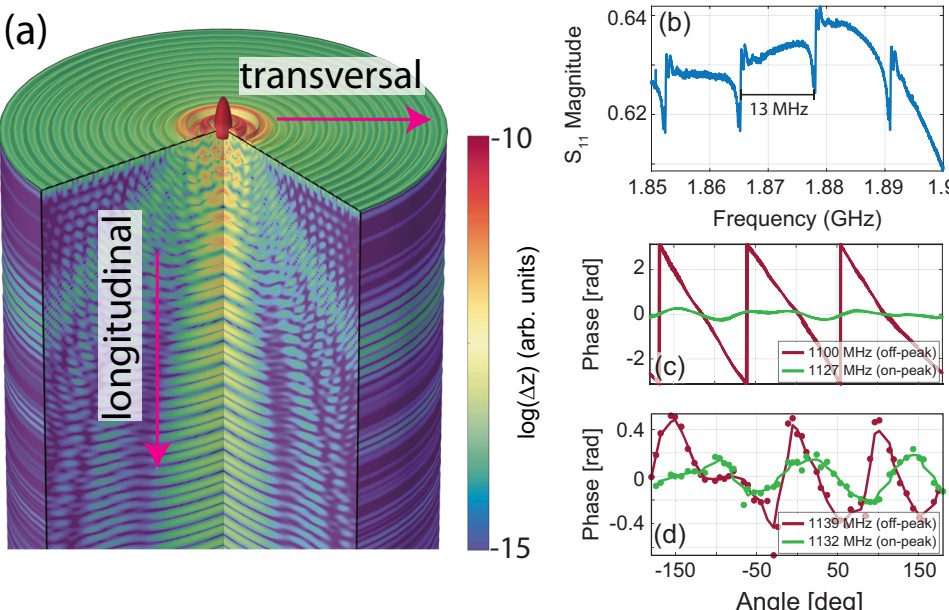

**Fig. 2 | Electrical response. a** Simulated mechanical displacement ($\Delta z$, in log-scale) originating from a circular BAWR in a 2D, azimuthally symmetric cell. **b** Experimental electrical reflection (corresponding to the $S_{11}$ rf-scattering parameters) of the BAWR in a limited frequency range, highlighting the 13 MHz separated dips corresponding to substrate resonances. **c** Simulated and **d** experimental phase profile around the path $\gamma_c$ of Fig. 1g, evaluated precisely at one of the substrate resonances in panel (**b**) (on-peak) and far from them (off-peak). The dots are experimental points, with the solid line representing an average. The phase excursion decreases at the resonance frequency in both simulation and experiment.

defining its outer rim as a single spiral loop of radius:

$$\text{AS}(\theta) = R_0 + g\frac{\theta}{2\pi}, \quad \text{with} \ \ \theta \in [0, 2\pi] \tag{2}$$

where $R_0$ is the spiral starting radius and $R_0 + g$ is its maximum radial extension.

The amplitude and phase maps of a typical vortex generated for an applied driving frequency of 1.14 GHz, are shown in Fig. 1d, e. The maps have been obtained using the 532-nm light from a solid state laser; the optical beam was focused on the device top surface and the amplitude and phase profiles analyzed with a Michelson interferometer coupled with to a vector-network analyzer for coherent detection of the complex field amplitude (more details can be found in the Methods section). The focused beam size was ~1 μm, making it smaller than the observed acoustic features at driving frequencies not exceeding 2 GHz. By sweeping the beam position, complex field maps can be reconstructed, which are similar to those delivered by dynamic digital holography, where larger field of views are employed for a one-shot evaluation of extended acoustic maps[40], see also Supplementary Note 8. Finite-element simulations of the structure of Fig. 1d, e within a simplified, albeit 3D toy-model (see Methods) well reproduce the observed feature, both for the amplitude [Fig. 1f] and phase [Fig. 1g]. The latter map reproduces qualitatively the same features observed in the experiments, although with a scaled range whose origin will become clear in the following discussion. A comparison of experimental and simulated vortex animation is shown as a Supplementary Video.

Interestingly, the observed vortex has a $C_3$ symmetry (i.e., a threefold periodicity along $\hat{\theta}$). This is linked to the presence of a background signal coming from the combination of the transversal and longitudinal components of the waves generated in our device.

The waves emitted by a circular BAWRs have, in fact, a prevalent longitudinal component along with a weaker transversal one (cf. simulation of Fig. 2a for a circular BAWR). The deformation of the circular BAWR into an AS contact engineers the wavefront of the transversal component, giving rise to the vortex. The longitudinal

component is weakly affected by the AS shape and, therefore, we can assume that the BAWR generates both irrotational waves ($\ell = 0$), linked to the longitudinal component, and rotating ones ($\ell \neq 0$) giving rise to the vortex. Additionally, the longitudinal component is further enhanced for frequencies resonating with the substrate thickness: multiple reflections at the substrate boundary result in sharp resonances, which can be clearly identified in the electrical radio-frequency reflection (S-parameter $S_{11}$) signal S-parameter reflection ($S_{11}$). Those have been observed also in our device, as illustrated by the sharp resonances in Fig. 2b and, owing to the nature of acousto-optic coupling, are related to the maxima of the interferometric signal. The peak-to-peak separation, of about 13 MHz, corresponds to the frequency spacing between longitudinal acoustic modes with a velocity of $10943 \pm 552$ m/s in the acoustic cavity defined by the boundaries of the c-oriented sapphire substrate with a nominal thickness of 425 μm. The estimated velocity is within the range reported for picosecond ultrasonic cryogenic measurements[41]. While the detailed description of the wave components is out of the scope of this manuscript and will be reported somewhere else, here we limit ourselves to mentioning that both longitudinal and transversal components are pervasively present at every excitation frequency, with a dominance of the former one at frequencies corresponding to the substrate resonances.

The longitudinal wave component acts as an oscillating background to the engineered planar wavefront which creates the vortex. A detailed model is reported in the Supplementary Note 2, here we just mention that the background modifies both the functional shape of the amplitude and phase signals. The phase excursion evaluated in a path around the vortex reduces with increasing background signal; the phase itself, which linearly grows from 0 to $2\pi\ell$ for a pure vortex of topological charge $\ell$, becomes a smoothly oscillating function. The amplitude evaluated along the same path undergoes a similar modification, becoming an oscillating function with a number of oscillations equal to $\ell$ in one revolution. This effect was numerically verified employing 3D simulations in a frequency range around the one used to characterize the vortex in Fig. 1. The phase evaluated within the dashed-line path of Fig. 1 for a frequency with negligible background contribution, i.e., away from the longitudinal wave resonances

(off-peak conditions), shows the expected roughly linear trend (red line in Fig. 2c), with a total $2\pi \times 3$ accumulated phase, thus evidencing the presence of a vortex with $\ell = 3$. Conversely, the phase evaluated precisely on-resonance shows a decreased excursion range while retaining the 3-peaks signature of a vortex with $\ell = 3$ (green line in Fig. 2c). Further simulations exploring the role of the substrate in contributing to the background signal are reported in the Supplementary Note 7, where we compare acoustic fields for perfectly or weakly reflecting substrate bottom face. The same qualitative behavior has been observed in the phase extracted from the experimental maps: as shown in Fig. 2d, the phase excursion is larger when the driving frequency is far from resonance (red curve), while it appears reduced for perfect resonant drive (green curve). Both dataset show the 3-peaks signature indicating the topological charge; the lack of quantitative agreement with the simulation has to be searched in the smaller substrate thicknesses used in the simulations, which modifies the resonant position and weights of the background signal.

The directly generated acoustic field is deeply intertwined with the optical field, as can be hinted by the characterization of mechanical fields through light interferometry. As one can guess, the main acousto-optic coupling mechanism relies on the differential path for the reflected light due to the moving boundary effect: imagining the device top surface to be illuminated by a planar wavefront, we expect that the reflected wave isophase surface strictly follows the out-of-plane displacement, see the sketch in Fig. 3a, b. The reflected wavefront can then be evaluated by interferometry, as done in our experiment, where the overall field is reconstructed by composing the measurements at different spot positions. Mathematically, the reflected electric field depends on the out-of-plane displacement $\Delta z$ as:

$$E_r(r, \theta) = \mathcal{R} \cdot E_i(r, \theta) \exp\left[2\pi i \frac{2\Re[\Delta z(r, \theta)]}{\lambda_o}\right] \quad (3)$$

where $\lambda_o$ is the optical wavelength, $\mathcal{R}$ the surface reflectivity, $E_r$ and $E_i$ the reflected and incident electric field amplitudes, respectively, and $\Re$ denotes the real part of the generally complex displacement $\Delta z$. Note that this coupling mechanism will be accompanied by a possible modification of the material refractive index due to the strain of the acoustic wave (elasto-optic effect). Since our device top contact is made of metal, we assume that the latter is negligible with respect to the former, which we will employ in the following discussion. We can describe the acoustic field at the device top surface, in a first approximation, employing modified Bessel's functions of the first kind (more details in Supplementary Section 2.B in the Supplementary Notes). Figure 3c reports amplitude and phase for the $\ell = 1$ acoustic vortex, where the maximum displacement amplitude has been taken as $\lambda_o/4$. Using Eq. (3), we can reconstruct the reflected optical field, where we assumed illumination with a planar wavefront ($E_i(r, \theta) = E_i$) and perfect reflectivity ($\mathcal{R} = 1$), resulting in the amplitude and phase field maps of Fig. 3d.

Noteworthy, while the reflected optical field carries an angular momentum, by rotating at the acoustic frequency, it is not an optical vortex, missing the structure of a proper vortex, including the central singular point with zero amplitude. Nevertheless, the OAM-carrying beams here obtained maintain the properties of a good basis, with states formed with different topological charge being orthogonal to each other. The orthogonality is a general property of vortices (see, for example ref. 42), which is in fact well verified for the acoustic fields. Figure 3e reports the dot product matrix (Gram matrix) of a generic acoustic vortex state $|\Psi(\ell_i)\rangle$, i.e. $G_{ij}^m = \langle \Psi(\ell_i)|\Psi(\ell_j)\rangle$ calculated as described in Supplementary Note 2.C. The matrix $G_{ij}^m$ is clearly diagonal, suggesting that the states (from $\ell = 1$ to $\ell = 3$ in this case) form an orthonormal basis.

The same matrix operation can be performed to determine the orthogonality properties of the obtained OAM-carrying optical beam, here defined as $|\psi(\ell_i)\rangle$, yielding the matrix $G_{ij}^o$ reported in Fig. 3f. Contrary to the acoustic case, the orthogonality for the optical beam depends critically on the ratio $\Delta z_M/\lambda_o$ between the maximum of the real part of the acoustic displacement amplitude ($\Delta z_M = \max[\Re(\Delta z)]$) and optical wavelength ($\lambda_o$). A quasi-orthonormal basis can be found for ratios exceeding ~ 0.15 with a small overlap between states carrying different $\ell$, as indicated by the small non-diagonal elements of $G_{ij}^o$ (i.e., with $i \neq j$, cf. Supplementary Note 2.C). We note, however, that these ratios are much larger than the values of ~0.1% for our present experimental conditions. They can, however, be significantly increased by using e.g., metasurfaces[17] or optical cavities[43] to enhance the acousto optical interaction. The $\lambda_o$-dependence favors, nevertheless, the use of the present approach in the context of ultraviolet vortex beams with no need to resort to high laser power and nonlinear processes, as customarily done[44]. Supplementary Note 2.C reports more detailed calculations of the modification of the acoustic and optical basis under different displacement amplitudes and contributions of the acoustic background, the latter weakly impacting the optical mode orthogonality.

## Vortex characterization

The possibility of generating vortices with different topological charges is highly desirable, enabling applications such as OAM-multiplexed telecommunication[21–23] and topological control of light[3,45] and vibrations[46,47]. The vortex phase profile can be controlled by engineering the radial pitch of the spiral arm, which defines the top contact shape. This is parameterized by the ratio $\eta = g/\lambda_a$, where $\lambda_a$ is the acoustic wavelength and $g$ the spiral spoke length, see Eq. (2). A vortex with a certain $\ell$ can be then obtained by employing a spiral-shaped BAWR with $\eta = \ell$. This was verified by performing FEM 3D simulations at a fixed frequency and varying $g$, which yielded indeed vortices with $\ell$ ranging 0 to 4 for a fixed driving frequency of 1.134 GHz, in agreement with what one would expect by evaluating $\eta$ (cf. Supplementary Note 3).

The geometrical modification translates into a smooth monotonic variation of the calculated out-of-plane angular momentum, suggesting the presence of very broad and overlapping resonances, each of them centered at $\eta = \ell$. This fact enables the creation of rotating fields in a continuous frequency range and facilitates broad-band modulation in our excitation range from 0.5 to 7 GHz. Broadband operation is granted by the specific geometry of our device, with a solid contact whose outermost rim is carved like a spiral; properly shaped BAWR, including multiple metallic fingers arranged in a spiral guise, would lead to sharper resonances and different regime of operations. This would, in fact, impose stricter in-plane resonant conditions, where the finger spacing would determine the acoustic wave generation, leading to higher purity vortices while reducing the frequency tunability of the topological charge. Furthermore, our simulation excludes the creation of fractionally-charged vortices, which can be observed in more complex configurations, see for example ref. 48.

In a more practical approach, the vortex topological charge can be tuned by keeping the same BAWR geometry (and, thus, $g$) and simply changing the driving frequency (and thus the acoustic wavelength $\lambda_a$). We verified this type of frequency-based $\ell$ tuning by experimentally evaluating vortex maps for selected frequencies in the range from 0.5 to 1.5 GHz. Representative measurements are reported in Fig. 4 and in the Supplementary Note 5. For the sake of measurement speed, all chosen frequencies were tuned to the strongest interferometric signal, which corresponds to a longitudinal BAW resonance, cf. Fig. 2b. Even under the strong background signal for the on-peak excitation, we can identify the vortex topology by looking at the number of phase oscillations in the path around the vortex center, as previously discussed. The experimental phase maps (**exp**) are compared with the simulated

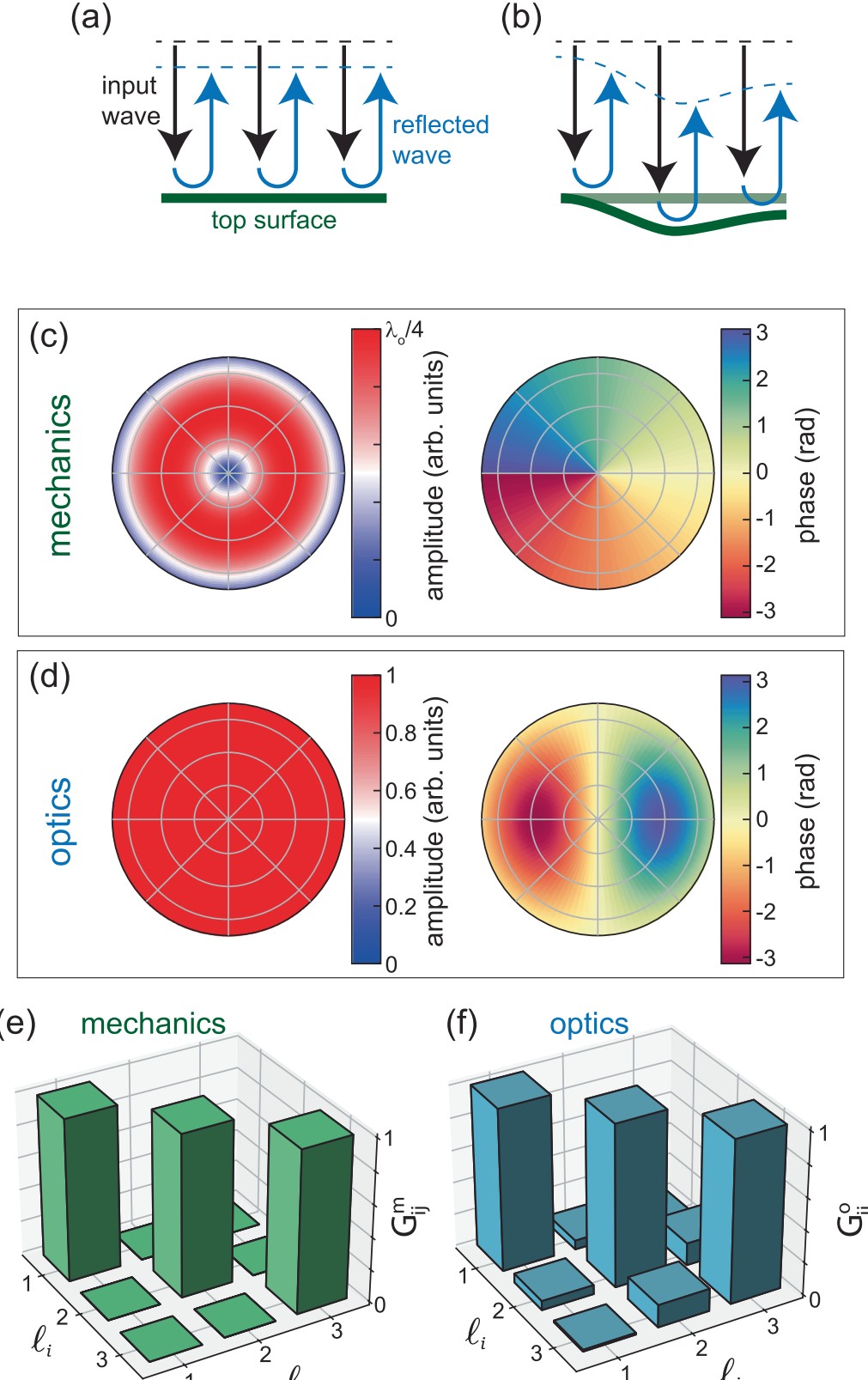

**Fig. 3 | Mechanical and optical angular momentum. a**, **b** Sketch of the moving boundary coupling mechanism, relying upon the phase shift of reflected waves. **c** Amplitude and phase polar maps of the acoustic vortex with $\ell = 1$. **d** Amplitude and phase polar maps of a perfectly reflected planar optical wavefront, considering a maximum acoustic displacement of $\lambda_o/4$. **e**, **f** Gram matrices of the mechanical and optical bases considering the range of $\ell$ from 1 to 3.

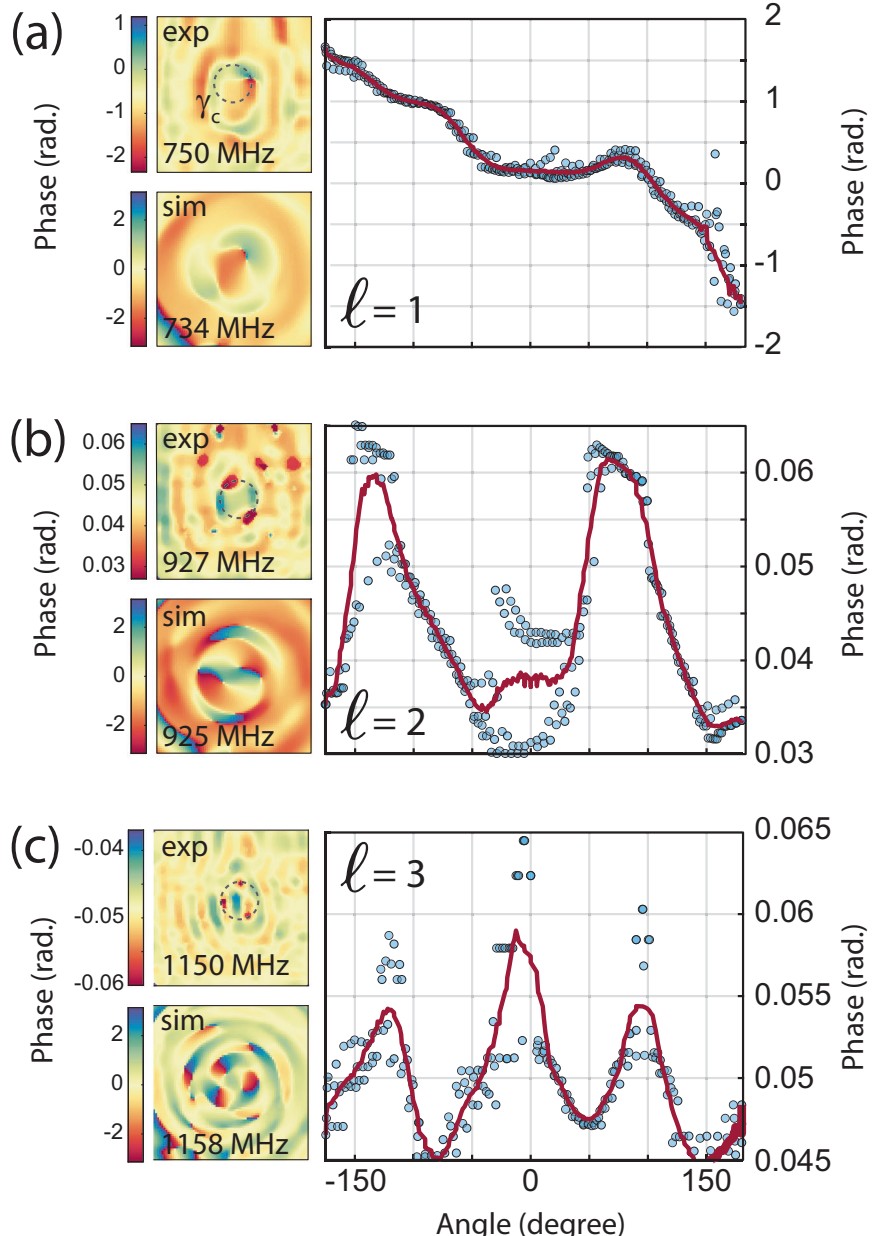

**Fig. 4 | Electrically tunable vortex topological charge. a** Phase profile around a vortex with topological charge $\ell = 1$ generated by driving the BAWR at 750 MHz. Experimental data points are the blue dots, with the red lines showing their moving mean value. The upper and lower left insets show the measured and simulated

phase profiles. The dashed circle in the upper left inset shows the circulation path for the determination of the phase profile. **b, c** Corresponding data for vortecies with $\ell = 2$ and $\ell = 3$ excited at 927 MHz and 1150 MHz, respectively.

ones (**sim**), showing a good qualitative agreement between to two for all driving frequencies. Following the previous discussion, we evaluated the experimental phase profile along $\gamma_c$, which has been indicated with a dashed circle in the figure maps. As expected, we found a monotonic phase change at 734 MHz, indicating a vortex with $\ell = 1$, see Fig. 4a. At higher frequencies, we found an increasing number of oscillations, namely two and three at 925 MHz and 1158 MHz, respectively. This indicates the presence of vortices with $\ell = 2$ and $\ell = 3$, respectively. Note the very different phase range in the experiment, as expected by the increasing strength of the longitudinal component background signal with frequency. The latter is compatible with the fact that we are exploring the low-frequency end of the BAWR generation band, which is centered around 4 GHz and roughly spans from 0.5 GHz to 7 GHz (More details can be found in the Supplementary Note 1). The spatial resolution of the interferometric setup (of ~1 μm)

limits the largest $\ell$ to 4 as measured at 1325 MHz, see Supplementary Note 4.

Further insights on the expected maximum topological charge has been gained by running simplified FEM simulations, where we considered a purely mechanical model with a single boundary excitation in the shape of the spiral contact of the BAWR. Despite the rough simplifications, the model well reproduces the main features observed in the experiment, while allowing us to run simulations at an increased spatial resolution. This translated to the observation of a vortex with $\ell = 13$ at 5 GHz excitation frequency, which represents a lower bound for the device capabilities.

Different experimental approaches could be employed for mapping acoustic vortices at very large frequency, such as acoustic atomic force microscopy[27]; alternatively, it is possible to get access to higher $\ell$'s within the same frequency range by realizing more complex spiral

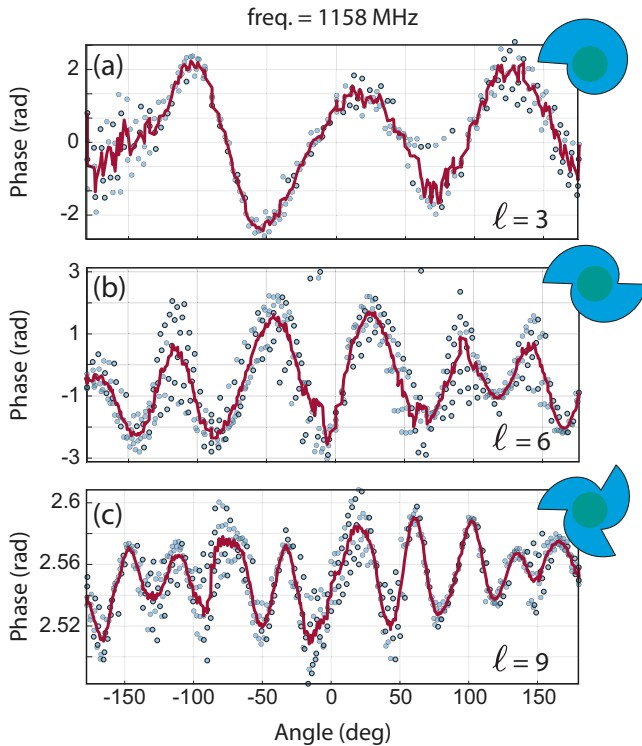

**Fig. 5 | High topological charge vortices.** Comparison of phase profiles for spiral BAWRs of different $M$s, operated at the same frequency of 1158 MHz. **a** $M = 1$ spiral, **b** $M = 2$ spiral, **c** $M = 3$ spiral. All spiral shapes have been sketched in the upper-right corner of their respective plots. Experimental data points are the blue dots, with the red lines showing their moving mean value.

contacts. A natural choice would be to consider Archimedean spirals with multiple spokes, which can be introduced by a simple modification of Eq. (2):

$$\mathrm{Sp}(\theta) = R_0 + \frac{\mathrm{mod}\ (M\theta, 2\pi)}{2\pi} \cdot g \qquad (4)$$

where mod is the modulo function and $M$ is an integer number defining the number of spokes in the spiral. This modification is expected to change the resonant condition for the generation of a vortex with topological charge $\ell$ as: $\eta' = M \times \eta = \ell$, with $M, \eta \in \mathbb{N}$.

This approach allows a simple scale-up of the topological charge at the same operating frequency by considering $M > 1$ in the spiral geometry. Measurements demonstrating this concept are reported in Fig. 5.

Figure 5a–c compares circulation phase profiles recorded by exciting BAWRs with an increasing number of spokes (cf. upper right inset) at a fixed frequency of 1158 MHz. The single-spoke spiral BAWR ($M = 1$, cf. Fig. 5a) generates a vortex with $\ell = 3$ for this excitation frequency. For the very same excitation frequency, the double- ($M = 2$) and triple-spoke ($M = 3$) spiral BAWRs generate, as expected, vortices with $\ell = 6$ and $\ell = 9$, respectively (cf. Fig. 5b, c). The Supplementary Note 6, reports the full 3D simulation of the devices with different number of spokes. The scaling-up effect is limited by the device lateral dimension; if $M$ is large enough to reduce the angular spoke-to-spoke distance to less than an acoustic wavelength, it will not allow the formation of the proper planar wavefront which generates the vortex.

The phase excursion around the vortices in Figs. 4 and 5 reduces with the acoustic frequency and with the increasing number of spokes. Off-peak excitation in-between the $s_{11}$ resonances (cf. Fig. 2b–d) would, of course, increases the observed phase excursion, as demonstrated in Fig. 1d, but simultaneously decreases the signal-to-noise level,

conversely increasing the measurement time. Note that working in different configurations, for example using a substrate with a non-polished bottom face could lead to a strong reduction of the resonant background peaks, as successfully demonstrated in ref. 49 and in Supplementary Note 7. Moreover, different BAWR designs, for example substituting the filled AS with single spiral fingers, would lead to a stronger resonant condition in the $\hat{r} - \hat{\theta}$ plane, enhancing the transversal, $\ell \neq 0$ wave component while suppressing the longitudinal, $\ell = 0$ one, the latter being the main contribution to the acoustic background, see more details in Supplementary Note 7.

In conclusion, we have experimentally demonstrated the generation of GHz acousto-optical vortices with tunable topological charge reaching up to 9 using a piezoelectric BAWR. The BAWR, which has a compact design based on a double-contacted piezoelectric layer, generates acoustic vortices propagating into a sapphire substrate. Simultaneously, the acousto-optical interaction at the top surface imparts a specific wavefront to light, creating time-varying optical beams with OAM modulated at the mechanical frequency. Using an interferometric setup we performed coherent characterization of the vortices, detecting both in-phase and quadrature signal yielding, therefore, access to both amplitude and phase of the acoustic field. The experimental studies are complemented by modeling and simulation of the acousto-optic waves. As a distinguishing feature of our device, we demonstrated the dynamic control of the acoustic vortex topological charge by simply tuning the excitation frequency, demonstrating a continuous tuning from $\ell = 1$ to $\ell = 4$, a maximum experimentally demonstrated $\ell = 9$ and estimating a lower bound of $\ell = 13$, limited by our simulation platform. The operating frequency of the acoustic vortices, reaching 7 GHz (with the potential of reaching 20 GHz in similar systems[50]), exceeds by more than one order of magnitude the largest ones reported in the literature (250 MHz[51]), while the GHz modulation speed of the time-varying OAM optical beams well compares with other reports of $\ell$ − tunable, solid-state generators (with modulation frequency of -MHz[52]).

The operating frequency range makes it suitable for a new generation of hybrid systems, where acoustic waves are combined with photonic and electronics quantum nanodevices sharing similar, micrometric-sized footprints[53–55]. On a very general ground, the device characteristics can unveil interesting applications in several fields: from angular-momentum based optical communications[3,4,56], where $\ell$-tuning and non-resonant light interaction are beneficial to applications, to topological acoustics[12] and acoustic tweezers[14,15], where it is possible to achieve chiral-based 4 degree of freedom particle manipulations[57] with the added features of operating at GHz frequencies, where accelerations as high as -$10^6 g$ have been predicted for microjets in fluids[58].

Simple design modifications, e.g., by replacing the full contact BAWR by spiral-like metallic fingers, would produce sharp resonances at different $\ell$s at expenses of tunability, yet offering a different approach tailored to specific applications where high-purity vortices are needed. Our platform offers a powerful tool for introducing angular momentum degree of freedoms in hybrid quantum systems, such as, for instance, optomechanical excitonic polaritons[28], where OAM quantization has been recently demonstrated in laser stirring experiments[59,60] as well as magnons[61] and magnetic color centers[62].

## Methods
### Device fabrication
Devices have been fabricated starting with a 425-μm thick, double-polished sapphire substrate. The BAWR bottom contact was defined using optical lithography followed by the sputter deposition of a ZnO piezoelectric layer and the photolithographic lift-off fabrication of the 10/30/10 nm of Ti/Au/Ti top contact. The bottom contact was shaped as a ring with external radius 25 μm and internal one 10 μm. The piezoelectric ZnO layer was sputtered at 150 °C with a thickness of 700 nm.

This layer has been subsequently patterned using optical lithography and reactive ion etching. The specific shape of the Archimedean spiral top contact was designed using $R_0 = 12.5\,\mu m$, $g = 16.67\,\mu m$ and different $M$'s for the devices with increased number of spokes, see Eq. (4).

## Numerical simulations

Numerical simulations were performed using a commercial finite-element method solver (Comsol Multiphysics). Fully coupled electrostatic and mechanical differential equation systems have been solved in order to use a radio-frequency voltage as input and evaluate the excited mechanical displacement. The two different simulations reported in the manuscript considered a 2D model with full azimuthal symmetry, nominal layer thickness for all the employed materials and a circular BAWR with radius $12.5\,\mu m$ (Fig. 2). Full, 3D simulations considered a $M = 1$ spiral with the same nominal dimensions as the fabricated device. Due to memory constrains, while the contacts and ZnO layers have been simulated using their nominal geometric values, the sapphire substrate was assumed to be acoustically isotropic with a thickness of $25\,\mu m$ (Figs. 1 and 3). While supporting different resonant conditions for the longitudinal substrate waves, this simplification did not change the qualitative results of the simulations, introducing only a possible mismatch due to the slightly different overlap between the vortex field and the longitudinal wave component.

## Experimental setup

The acousto-optical device characterization has been performed using a Michelson-Morley interferometer mounted on a movable microscope head. The device was excited using radio-frequency probes coupled to the port 1 of a Vector Network Analyzer (VNA). A continuous-wave laser light at 532 nm was split into two arms, one ending at an oscillating mirror (modulated at low frequency $f_m$) mounted on a piezoelectric actuator (stabilization arm) and the other leading to a microscope head pointing at the device top surface, which focuses the light beam onto a spot size of around $1\,\mu m$. The optical beam was circularly polarized, even if we do not expect any substantial dichroism effect given by the small beam size when compared with the acoustic features under investigation. Once the two beams were recombined upon reflection, the interferometric signal was fed to a fast photodetector. The slow monitor signal at $2f_m$ was used to stabilize the interferometer working point by adjusting the position of the mirror actuator in the stabilization arm; conversely, the fast AC signal recorded by the photodetector was fed to port 2 of the VNA, leading to a spectrally and phase-resolved resolved detection of the surface displacement profile through the evaluation of the $S_{21}$ scattering parameter. The field map reconstruction via beam spot scanning yields results equivalent to those obtained using wide-area illumination, as employed in dynamic digital holographic microscopy for measuring low-frequency vibrations. While the latter technique is convenient for rapid characterization - since it does not require scanning the beam position - it would necessitate highly sophisticated detectors, specifically cameras with a large number of pixels and sub-GHz frame rates, to be suitable for our experiment

## Data availability

Raw data, data analysis, and plotting scripts are available in the Zenode repository under accession code https://doi.org/10.5281/zenodo.14512810.

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

## Acknowledgements

A.P. acknowledges funding from the Alexander von Humboldt Stiftung through the "Experienced Researchers" fellowship program, PVS acknowledges funding from the German DFG (grant #426728819). We also thank A. Riaud and M. Baudoin for discussion in the initial phase of these studies.

## Author contributions

A.P. and P.V.S. conceived and designed the experiment. A.P. and I.d.P.E. characterized the device electrically while A.P., N.A., M.M. and P.V.S. performed the optical interferometric measurements. A.P. and P.V.S. discussed the results, carried out the modeling and performed the numerical simulations. A.P. wrote the manuscript with input and discussion from all the authors.

## Funding

## Competing interests

The authors declare no competing interests.
