## [Transparent Peer Review file · Nature Communications]

GHz acousto-optic angular momentum with tunable topological charge

Corresponding Author: Dr Paulo Santos

Version 0:

Reviewer comments:

Reviewer #1

(Remarks to the Author)

The authors present a method for inducing vortex acoustic beams in solid media, using a spiral structure with piezoelectric materials to generate acoustic vibrations with non-zero topological charges. By adjusting the operating frequencies or altering the spiral geometry, they demonstrate the generation of vortices with high topological charges.

Firstly, I have some concerns about the clarity of expressions and presentation. The authors claim that their approach can generate time-varying OAM optical beams; however, I did not find any experimental figure demonstrating these generation results. I am uncertain whether this is an assumption, or a derivation based on the measured vibration pattern rather than an actual experimental measurement.

The paper only references an optical wave in the form of a focused laser, used to measure surface mechanical vibrations. Employing a focused laser to scan surface vibrations is a well-established, commercially viable technique (e.g., 3D laser vibrometry). However, measuring a vortex vibration pattern differs from generating a chiral optical beam. The radius of a vortex beam at the laser wavelength would be smaller than the radius shown in Fig. 1.

The authors mention the acousto-optical generation of chiral optical beams. However, the only material capable of modulating permittivity is a ZnO piezoelectric layer, which is covered by a metallic layer on top. This metallic layer would reflect the incident laser light, preventing any interaction between the light and the piezoelectric layer. I am therefore uncertain as to whether the acousto-optical interaction could occur in this setup and facilitate the generation of chiral optical beams.

Besides, there are some technical concerns in the following:

1. Could the authors further optimize the spiral structure to improve the quality of the acoustic OAM beam? Figures 1, 3, and S3 indicate an unavoidable background that distorts the wavefront, suggesting that the current design could still be optimized. I believe it would be beneficial to discuss methods for removing or suppressing this background contribution.

2. In Figs. 3(b,c) and 4(c), the plot legends and ranges for the experimental results differ significantly from those in the simulations. If the phase were plotted over the same range as in the simulations (i.e., $-\pi \sim \pi$), it may not sufficiently confirm the observation of vortex beams with high topological charges. Improvements in the experimental measurements and the consistency of these representations would strengthen the claim.

To summarize, I cannot recommend the publication in the present form. A careful revision is required, and some statements should be checked.

Reviewer #2

(Remarks to the Author)

A. Pitanti et al. present a novel acousto-optical vortex beam generator based on a piezoelectric bulk acoustic wave resonator (BAWR). This device can generate acoustic vortices with frequencies around 4 GHz on a solid-state substrate and

offers tunable topological charges from 1 to 13 or higher. By changing the driving frequency, the topological charge of the acousto-optical vortices can be externally controlled within a single device, providing new solutions for light communication based on angular momentum, 3D particle manipulation, and novel optomechanical device interactions. As an innovative acousto-optical coupling tunable OAM generator, I believe it has the potential to be published in Nature Communications, provided the authors address the following questions:

1. In Fig. 1, "Acousto-optical generation of chiral optical beams," is the term "chiral optical beams" accurate when only OAM-carrying beams are involved?
2. Is the scale of Fig. 1(b-c) consistent with that of (d-f)? Please add a scale bar.
3. The statement "a standard circular launcher, in fact, creates a radial wavefront which can be described," together with Fig. 1(b-c), implies a radially polarized beam is generated. Which component of the electric field is represented in the phase shown in Fig. 1(e)(g)?
4. What are the specific frequencies for "on resonance" and "out of resonance" in Fig. 2(c-d)? There appears to be a peak near 0 degrees in (c)—what is the cause of this?
5. The inset of Fig. 3 requires a scale bar. The experimental difference in Fig. 3(b-c) is only approximately 0.03 (rad), indicating low resolution. According to the authors' theory (Equation 1 in the supplementary material), the background signal's (longitudinal component) amplitude is an order of magnitude larger than the vortex signal (transverse component). How is the circulation path chosen, especially in Fig. 3(c) where there is a significant difference compared to simulations? Are there any solutions to reduce the background signal amplitude?
6. In the supplementary material, does Fig. 2(a-b) correspond to cases where the propagation axes of the acoustic and optical waves are aligned and misaligned, respectively?
7. In Equation 1 of the supplementary material, the background amplitude (Bkg) only significantly affects phase amplitude when it is around 1. When Bkg is much less than 1, it has minimal impact on the overall amplitude and phase; when Bkg is much greater than 1 (as in Question 5), the phase change becomes negligible. What criteria did the authors use to choose the value of Bkg, such as the 0-2 range selected in the supplementary material?
8. On page 3 of the supplementary material: "In a first approximation, a circular BAWR can be approximated as as a drum resonator"—the phrase "as as" is redundant.
9. In Equation (2) of the supplementary material, there is no description of the constant C.
10. The supplementary material's Fig. 3 lacks figure numbers, although the caption includes (a) and (b), and no scale bar is present. The term "gs" is not explained. Fig. 3 shows that as B increases from 0 to 1, the phase singularity shifts from the center. For $l=2$, it appears that a single OAM-carrying ring beam ($B=0$) splits into two $l=1$ beams ($B=1$). Although the total beam angular momentum remains 2, can it still be used for multiplexing in communication? I am curious about the effects when $B=2.5, 5, \text{ or } 10$.
11. The authors should provide simulation data for $l=6$ and $l=9$ in the supplementary material.
12. The authors may discuss and include more potential applications of such ultra-degree-of-freedom modulated vortex structured light such as ultra-capacity communications.

Reviewer #3

(Remarks to the Author)

The authors have experimentally demonstrated the possibility to generate acoustic vortices with tunable topological charges up to 9 in a wide frequency band around 4 GHz on the basis of a single-contact piezoelectric bulk acoustic wave resonator. The resonator is composed of a ZnO piezoelectric layer sandwiched between metallic contacts and it excites acoustic vortices in a sapphire substrate. The acoustic topologically charged waves are generated due to the fact that the resonator contact is modified into the shape of an Archimedean Spiral, which provides the required phase increment along a closed path around the center of the field. The vortex topological charge is then tuned by changing the driving frequency of the applied electric field. Besides, the generation of different acoustic topological charges was realized by making use of Archimedean spirals with multiple spokes at the fixed driving frequency. The experimental results were supported by numerical simulations with the use of commercial finite-element method solver (Comsol Multiphysics).

The authors claim that the presented device offers a new class of acousto-optic devices for generating optical vortices in the reflected beam from a normally incident Gaussian laser beam via acousto-optic interaction at the top surface. At that, they have not provided any measured data of the conversion efficiency of the incident optical beam into the produced optical vortex as well as of the generated optical intensity and phase distribution, and the corresponding topological charge. Also, nothing is said on the parameters of optical beams (beam waist, state of polarization, paraxial/non-paraxial). Even more importantly, the very description of acousto-optic interaction between acoustic and optical beams in the proposed scheme is also not presented in the manuscript. It is a critical drawback of the work because the physical process of the acousto-optic induced reflection accompanied by the formation of a vortex structure and transfer of OAM from longitudinally vibrating interface to the incident photons is far from trivial. In this connection, a number of fundamental questions arise:

1. What is the regime of the acousto-optic diffraction (Bragg or Raman-Nath)?
2. How do the kinematic and dynamic conditions look like for an efficient coupling of the incident and diffracted beams?
3. What is the dependence of the intensity of diffracted vortex beams on the waist of the incident beam, on the ratio of optical and acoustic wavelengths, on the state of incident polarization, and on the acoustic topological charge?

Since the efficiency of the acousto-optic diffraction critically depends on the interaction length and the sapphire substrate has a rather small thickness of 425 μm , it is reasonable to assume that the energy efficiency of the device should be rather low that undermines the practical value of the proposed scheme, especially allowing for a number of the previously demonstrated highly efficient methods of optical vortex producing. Besides, some of these approaches also allow for a

dynamic electronic control of the generated OAM, including those based on liquid-crystal spatial light modulators [A. Forbes et al., "Creation and detection of optical modes with spatial light modulators," *Adv. Opt. Photon.* 8, 200-227 (2016)] or acousto-optic interaction in optical fibers [M. Yavorsky et al., "Photon–phonon spin–orbit interaction in optical fibers," *Optica* 8, 638-641 (2021); Jiafeng L. et al., "Recent progress of dynamic mode manipulation via acousto-optic interactions in few-mode fiber lasers: mechanism, device and applications", *Nanophotonics*, vol. 10, 2021, pp. 983-1010].

As a second critical comment, it is stated in the text that the proposed approach allows one to create optical vortices with time-varying orbital angular momentum. I believe this is an erroneous conclusion drawn from a misunderstanding of the essence of the concept of "time-varying optical vortices (as well as spatio-temporal vortex beams)". In fact, these are the light pulses (polychromatic!) with the well-defined OAM that is changing in the space-time domain due to the inherent complex evolution of spectral components of the wave-packet [L. Rego et al., "Generation of extreme-ultraviolet beams with time-varying orbital angular momentum, *Science*, 364, (2019); Konstantin Y. Bliokh, "Spatiotemporal Vortex Pulses: Angular Momenta and Spin-Orbit Interaction", *Phys. Rev. Lett.* 126, 243601 (2021)]. Instead, like any other acousto-optic device, the proposed in the manuscript scheme simply enables switching between generated monochromatic optical vortices with different values of their fixed OAM by means of varying the applied driving frequency.

To summarize, the used in the paper method of generating acoustic vortices has been previously suggested and already experimentally realized, the acousto-optically-induced generation of optical vortices is totally not described, and creation of the time-varying optical vortices seems to be miss-attributed. From these perspectives, for the present paper, there is no novel finding and enough impact for publication in *Nature Communication*.

Version 1:

Reviewer comments:

Reviewer #1

(Remarks to the Author)

Thanks for the authors' clarifications and efforts for answering my concerns.

It seems that the authors adopt a broader definition of acoustic-optic interaction, whereas I prefer to view it as a type of boundary deformation effect. Under the authors' definition, nearly all natural materials could exhibit "acoustic-optic interaction". Since in general, optical waves reflect at boundaries or interfaces, and any boundary deformation influences the reflection phase. It would be reasonable if the authors explicitly stated that they are employing a generalized acoustic-optic interaction before using the term. However, I still recommend that they consider an alternative term to describe the mechanism in their paper to avoid potential confusion for readers.

The primary concern remains whether a vortex of surface out-of-plane vibrations can truly induce a vortex (or orbital angular momentum, OAM) in a reflected optical beam. First, note that generating a reflected optical beam with OAM is relatively straightforward. For example, by fabricating spiral structures (e.g., q-plates) on a metal surface, one can impose the desired phase on the incident beam, thereby producing a reflected OAM beam. Similarly, exciting a surface vortex is not particularly challenging, as previous studies have demonstrated that spiral nanostructures can generate plasmonic vortices carrying OAM — a concept that should also apply to elastic waves. One key novelty of this work, in my view, lies in achieving this effect dynamically or through electrical tunability. Accomplishing this in a micrometer-scale structure is far from trivial and represents a significant challenge.

However, as mentioned in my previous response, for this primary concern, the authors have not yet provided any direct experimental observation of an optical vortex (or OAM). Instead, they measure surface vibrations by focusing a beam on each position of the metal plate and infer the presence of OAM in the reflected optical beam using the Huygens–Fresnel principle, as shown in Fig. 3. Considering this as an experimental study, I remain skeptical about whether it can truly generate OAM in an optical beam in practice.

Firstly, the derivation based on indirect experimental measurements looks not entirely solid. Regarding the "phase reduction" effect (see R1.5), I sincerely appreciate the authors' response that the substrate significantly alters the phase distribution, leading to substantial discrepancies between simulation and experimental data. However, the authors acknowledge that no improvements can fully eliminate this background effect. This raises a critical issue: while simulations suggest the possibility of generating an optical vortex, the experimental data deviate significantly from these predictions. How can we confidently conclude the presence of an optical vortex based solely on simulations? Moreover, the measured reflected phase shows only minimal variation — e.g., 0.03 rad in Fig. 4(b) and 0.02 rad in Fig. 4(c). Such a small phase difference seems insufficient to induce a clearly defined optical OAM beam.

Secondly, the experimental setup in this work is essentially a home-made laser vibrometer designed for micrometer-scale structures. As we know, a laser vibrometer measures surface vibrations by detecting Doppler shifts in the reflected laser beam caused by surface displacement. Vortex patterns similar to those presented by the authors are often observed in surface vibration measurements, particularly in wave interference from multiple scatterings or metamaterials — though typically at the millimeter scale. However, while these vortex patterns are detected optically, they correspond to vortices in the elastic field rather than optical vortices. This distinction is crucial, as the experimental setup is mainly designed for spectrally and phase-resolved detection of surface displacement rather than the direct observation of optical OAM. Detecting

OAM should be achieved through independent measurements, such as Fork grating interferometry.

To summarize, can the authors conduct an experiment that directly measures the optical vortex (or OAM)? If the authors can provide such results, I think it would effectively address technical concerns regarding this aspect of the study.

Reviewer #2

(Remarks to the Author)

The revised version is much improved, I believe it can be published now

Reviewer #3

(Remarks to the Author)

In my opinion the revised manuscript accompanied by the more detailed Supplemental Material presents the reported results in a much more clear and convincing way, in particular, giving a possible explanation of the origin of the observed acousto-optic coupling as well as addressing the obvious limitations of the proposed device.

Although the observed energy efficiency of acousto-optic-induced generation of OAM-beams is relatively small, the demonstrated bulk acoustic wave resonator, which is capable of generating acoustic vortices with widely tunable topological charges at previously non-reported GHz frequencies, will be of importance for future studies of photon-phonon, magnon-phonon, electron-phonon, and other interactions in quite different physical systems in photonics, spintronics, plasmonics, etc. So now I can recommend the manuscript for publication.

Version 2:

Reviewer comments:

Reviewer #1

(Remarks to the Author)

I have no further technical comments and can recommend the paper for publication now.

But before final acceptance, please address the following formatting issues:

1. The "Merged File containing manuscript text" does not incorporate all tracked changes (highlighted in red) from the "Article File with Track Changes". There appear to be processing errors, as some missing changes are simply marked with a "1". Please ensure the final text file is generated accurately with all revisions included.
2. On Page 7 of the "Article File with Track Changes", the final paragraph (highlighted in red) before the conclusion section contains missing references, indicated by an "?" symbol (As mentioned above, this paragraph is not included in the "Merged File containing manuscript text"). Please verify and complete the references accordingly

REVIEWERS COMMENTS

Reviewer #1 (Remarks to the Author):

The authors present a method for inducing vortex acoustic beams in solid media, using a spiral structure with piezoelectric materials to generate acoustic vibrations with non-zero topological charges. By adjusting the operating frequencies or altering the spiral geometry, they demonstrate the generation of vortices with high topological charges.

We thank Reviewer #1 for their evaluation of the paper, suggesting important and necessary revisions to our manuscript.

C1.1: Firstly, I have some concerns about the clarity of expressions and presentation. The authors claim that their approach can generate time-varying OAM optical beams; however, I did not find any experimental figure demonstrating these generation results. I am uncertain whether this is an assumption, or a derivation based on the measured vibration pattern rather than an actual experimental measurement.

R1.1: We acknowledge that our statement about the generation of optical beams with time-varying OAM was not properly explained in the manuscript, especially considering the lack of a dedicated figure or discussion. Extrapolating from our experimental findings, we have now added more material to the revised version, including a detailed description of the structured optical beams that would be generated upon acousto-optic interaction. In particular we describe their orthogonality for different acoustic driving and at varying level of acoustic background, generalizing the discussion from our specific device to future experiments and improved realizations. The new data clearly shows that acousto-optic interaction can deliver angular momentum -carrying optical beams, whose helical phase is oscillating at the mechanical frequency; depending on the amplitude of the mechanical motion, the OAM-carrying beams show a strong orthogonality, fulfilling one of the conditions to use them as a base for light-based communications.

Changes in the draft: we inserted a new paragraph under Device Concept (starting with “The directly generated acoustic field ...”) and included a new figure (Fig. 3) in the main text. In addition, we added a new subsection (II.C) to the supplementary material including 2 new figures (Fig. S4 and Fig. S5).

C1.2: The paper only references an optical wave in the form of a focused laser, used to measure surface mechanical vibrations. Employing a focused laser to scan surface vibrations is a well-established, commercially viable technique (e.g., 3D laser vibrometry). However, measuring a vortex vibration pattern differs from generating a chiral optical beam. The radius of a vortex beam at the laser wavelength would be smaller than the radius shown in Fig. 1.

R1.2: We agree with the reviewer that measuring a vortex vibration pattern differs from generating a chiral optical beam. Conversely, our interferometric setup, which relies on a focused laser spot, enables the reconstruction of the phase map of the reflected light by scanning the illumination position. This approach yields results equivalent to those obtained with full-field illumination in dynamic digital holographic microscopes.

The main advantages of using a focused beam are that (i) the spatial resolution is inherently set by the beam size (ultimately limited by diffraction) and that (ii) it enables the use of fast detectors widely available with common technology. In contrast, while interference approaches using

wide-field illumination significantly reduce the acquisition time by eliminating the need for beam spot scanning, they require highly sophisticated detector systems (essentially cameras with a high pixel count and extremely high frame rates, reaching sub-nanosecond timescales in our experiment).

Our technique is well suited for measuring the rotational features observed in our experiment (up to a driving frequency of 2 GHz), since they are larger than the beam spot size ($\sim 1 \mu\text{m}$). Scanning the beam position allows us to fully infer the phase front of the reflected light, which would be enriched by orbital angular momentum (OAM) if a wider illuminating beam were used. Note that, as discussed in the following (R1.3) and the answers to Reviewer #2 and #3, the reflected light is not a proper optical vortex, although a beam carrying OAM with orthogonal properties when different topological charges are excited.

Changes in the draft: We reworded and expanded few paragraphs in the sections Device Concept (starting with “The focused beam size...”), Vortex Characterization (starting with “The directly generated acoustic field...”) and Methods – Experimental setup (starting with “The optical beam...” and “The field map reconstruction...”).

C1.3: The authors mention the acousto-optical generation of chiral optical beams. However, the only material capable of modulating permittivity is a ZnO piezoelectric layer, which is covered by a metallic layer on top. This metallic layer would reflect the incident laser light, preventing any interaction between the light and the piezoelectric layer. I am therefore uncertain as to whether the acousto-optical interaction could occur in this setup and facilitate the generation of chiral optical beams.

R1.3: Regarding the coupling mechanism: while resonant interaction within the material can indeed lead to an interesting acousto-optic coupling - as routinely achieved, for example, in optomechanics with optical fibers - we believe that in our device, light is primarily affected by vibrations via a purely geometric mechanism, i.e., through the phase shift induced by the differential path of reflected light following the out-of-plane mechanical displacement (see Fig. 3(a) of the manuscript, reproduced below). The coupling happens only at the topmost surface and it is favoured by a strongly reflecting layer, such as a metal, which is not, in fact, detrimental to the acousto-optic interaction. The photo-elastic coupling - i.e., the local changes in the refractive index induced by the strain field of the wave - is present but appears to have a secondary role. For the sake of discussion, we will neglect this second contribution which would otherwise improve our effective acousto-optic coupling.

The interaction here described is an effective and well assessed coupling effect, although we acknowledge that we failed at clearly describing it on the previous version of the manuscript, as well as underlining the required amplitude and polarization of the acoustic vibrations necessary for a strong coupling (as an example, lateral displacement negligibly changes the reflected light wavefront and is therefore essentially invisible to our technique).

Figure 3 of main text: (only panels (a) and (b)) Sketch of the coupling mechanism between optical wave and mechanical displacement

Changes in the draft: we added one new figure in the main text (Fig. 3) depicting the coupling mechanism, which is now discussed in a dedicated paragraph at the end of section Device Concept.

Besides, there are some technical concerns in the following:

C1.4: Could the authors further optimize the spiral structure to improve the quality of the acoustic OAM beam? Figures 1, 3, and S3 indicate an unavoidable background that distorts the wavefront, suggesting that the current design could still be optimized. I believe it would be beneficial to discuss methods for removing or suppressing this background contribution.

R1.4: The homogeneous spiral structure on doubly polished sapphire used for this experiment was designed aiming at having a broad and continuous tunability of the acoustic vortex topological charge along with strong vertical displacement for the modes resonating within the substrate thickness, favoured by strong reflection on the chip back facet. Indeed, while fulfilling the requirements described above, this design compromises with the presence of a strong, continuous background, which partially masks the generated vortices. This is essentially composed by the simultaneous generation of rotational ($\ell \neq 0$) together with irrotational waves ($\ell = 0$). Irrotational waves are essentially directed into the substrate and can form resonant modes within it, manifesting as sharp peaks as the ones reported in Fig. S7 in the new version of the supplementary information. Aiming at background-less modes, one can consider, as a first step, using a single-polished sapphire wafer, where bulk acoustic waves can be created and launched into the substrate with negligible back-reflection, as successfully demonstrated in [1]. While this approach would reduce the strong resonant peaks, further modification towards high purity vortices can be achieved by imposing a stronger on-plane resonant condition on the device geometry. For instance, using “single-finger” spirals instead of our current wide-area filled contacts, where only the outer rim is shaped as a spiral, could enhance this effect. This approach is expected to inhibit the generation of irrotational waves while enhancing the rotational ones, when the in-plane wavelength matches the finger spacing. This is shown in the preliminary simulation in Fig. R1 of this reply letter and in the attached “response_video.gif”; when using the proper resonant frequency, a clean vortex can be created, of course at expenses of the wide tunability we have shown in our experimental results.

We believe the different designs can be tailored to specific applications and adapted to one’s research goals.

Figure R1: FEM simulation (top-view) showing the real part of the complex displacement field for a “single-finger” spiral (dark red). The central acoustic vortex has topological charge = 1

Changes in the draft: we added a sentence in the Vortex Characterization section (starting with “Operating between peaks...”) and a sentence in the Conclusions (starting with “Simple design modifications...”) to assess possible device modifications tailored to obtaining background-less and high-purity vortices.

C1.5: In Figs. 3(b,c) and 4(c), the plot legends and ranges for the experimental results differ significantly from those in the simulations. If the phase were plotted over the same range as in the simulations (i.e., $-\pi \sim \pi$), it may not sufficiently confirm the observation of vortex beams with high topological charges. Improvements in the experimental measurements and the consistency of these representations would strengthen the claim.

R1.5: The reviewer is right in observing that the phase range in the simulation and experimental data differ significantly from each other. The experimentally observed “phase reduction” effect has been explained both in the manuscript draft and supplementary information: in short, we attribute that to the presence of a strong, non-rotational background component arising from the excitation of non-rotational waves ($\ell=0$), which can also be enhanced by resonating within the substrate. Due to the large substrate thickness, the resonances are closely spaced in frequency, making the background component essentially omnipresent. Moreover, its intensity gets stronger on top of a resonant peak and closer to the centre of the generation band of the BAW resonator (BAWR) (~ 4 GHz). We believe that no improvements in the experimental measurements can lead to an elimination of the background which would require instead a different device arrangement (i.e. a single polished material substrate).

Furthermore, the finite-element simulations have been performed on a toy model, where the substrate thickness has been greatly reduced in order to fit the simulation cell within the finite RAM of the computer. This results in widely spaced longitudinal resonances, which are not necessarily resonant at the simulated frequencies, which correspond to the ones of the experimental measurement. As can be seen this does not prevent the evaluation of the displacement field which is in good qualitative agreement with the experimental data, apart from the varying background contribution which modifies the overall phase range.

To demonstrate our hypothesis regarding the role of reflections at the backside of the substrate, we ran a further 3D simulation with a highly reflective back facet as in our experiment (fixed boundary), see Fig. S10 (a) (also reproduced below). This simulation is performed at a frequency

(1.121 GHz) resonant with one of the background peaks ($f=1.121$ GHz). As can be seen in the phase map, an acoustic vortex with topological charge 3 is found although with a strongly reduced phase range due to the presence of the background, confirming our findings in the main text. The very same simulation is run in a substrate with a weakly reflective back facet (i.e., by applying Perfectly Matched Boundaries - PML), see Fig. 4 (b). Decreasing the reflection strongly hinders the formation of substrate modes and, therefore, of the background signal. Under this condition, the phase range of the helical acoustic field becomes restored to the whole phase range, going from $-\pi$ to π , as can be seen in the Figure.

We believe that the current view of Fig. 4 in the main text is the best representation of simulation and experimental data; following the Reviewer’s suggestion and improving our description of the effect, we have added a sentence highlighting the different range at which the phases have been plotted and a new Figure (**Fig. S10**, corresponding to Fig. 4 shown above) and relevant discussion in the Supplementary Note 7.

Changes in the draft: we added a dedicated Figure in the Supplementary Material (**Fig. S10**); to further clarify the comparison between simulation and experiment we added a few sentences in the Vortex Characterization section (starting with “Operating between peaks...”).

To summarize, I cannot recommend the publication in the present form. A careful revision is required, and some statements should be checked.

We thank the reviewer for the constructive criticism of our draft which allowed us to identify weak points which we have amended in the current version of the manuscript. We hope that the current form will be satisfactory to the reviewer.

Reviewer #2 (Remarks to the Author):

A. Pitanti et al. present a novel acousto-optical vortex beam generator based on a piezoelectric bulk acoustic wave resonator (BAWR). This device can generate acoustic vortices with frequencies around 4 GHz on a solid-state substrate and offers tunable topological charges from 1 to 13 or higher. By changing the driving frequency, the topological charge of the acousto-optical vortices can be externally controlled within a single device, providing new solutions for light communication based on angular momentum, 3D particle manipulation, and novel

optomechanical device interactions. As an innovative acousto-optical coupling tunable OAM generator, I believe it has the potential to be published in Nature Communications, provided the authors address the following questions:

We thank the reviewer for their fair assessment of our manuscript draft and for suggesting that our research has the potential to be published in Nature Communications, once some issues have been properly addressed.

C2.1: In Fig. 1, “Acousto-optical generation of chiral optical beams,” is the term “chiral optical beams” accurate when only OAM-carrying beams are involved?

R2.6: The reviewer is right in pointing out this mistake. A vortex carrying OAM is not necessarily chiral, since chirality depends on a broken symmetry which would require spin-orbit coupling or the presence of a symmetry-breaking medium.

Changes in the draft: we removed the word chiral from the caption of Figure 1 and in another instance in the text.

C2.2. Is the scale of Fig. 1(b-c) consistent with that of (d-f)? Please add a scale bar.

R2.2: The maps shown in Fig. 1 (b-c) have been generated analytically starting from the well-known expression of the Bessel’s functions of first kind, which are solutions of the wave equation for a vibrating drum and well approximate the waves generated in-plane by a circular bulk acoustic wave resonator. Map (b) additionally considers a “brute-force” modification of the wavefronts adding a spiral extra-phase term. Both maps have just an illustrative purpose, to guide the reader through the reasoning behind our design and are not directly related to the experiment and device simulation of panels (d-f). Anyway, for the sake of clarity, we added a colorbar and defined the plotted quantity (amplitude) in both panels.

Changes in the draft: we **modified Figure 1** adding the colorbar where requested.

C2.3: The statement “a standard circular launcher, in fact, creates a radial wavefront which can be described,” together with Fig. 1(b-c), implies a radially polarized beam is generated. Which component of the electric field is represented in the phase shown in Fig. 1(e)(g)?

R2.3: We thank the reviewer for raising this point, which contributes to improving our explanation of the experimental results. A standard circular BAW also launches a radial acoustic wavefront with a strong out-of-plane polarization component. The later enables the direct mapping the acoustic field using light interferometry. The precise mode representation is, indeed, a little more complex and involves high order Lamb waves, whose description is out of the scope of this manuscript and will be reported in a dedicated publication.

The light electric field used to map the vibrations impinges perpendicularly to the top surface and is circularly polarized. The mapping is done by scanning the position of the laser beam on the device top surface. The beam spot has a size around 1 μm and it is, therefore, small compared to the acoustic features and wavelength. Additional details on our characterization technique can be found in the relative answer to Reviewer #1.

The local interaction of the acoustic and optical fields relies on the dynamical phase shift of light due to the out-of-plane vibrations (see Fig. 1 and the text replying to Reviewer #1) or photo-elastic coupling, therefore we assume it is not polarization-dependent in the local coupling approximation and considering materials (metals, ZnO and sapphire) which are optically isotropic on the top surface plane.

Changes in the draft: we added some sentences both in the Device Concept (starting with “The waves emitted...”) and in Methods – Experimental Setup sections to better clarify the measurement conditions, specifying the properties of the probing interferometric beam. See some of the precise modifications in our reply **R1.2**.

C2.4: What are the specific frequencies for “on resonance” and “out of resonance” in Fig. 2(c-d)? There appears to be a peak near 0 degrees in (c)—what is the cause of this?

R2.4: We understand the necessity of including this information in Figure 2. We added the experimental and simulated frequencies on peak (1132 MHz and 1121 MHz, respectively – labelled as **on**, from on-peak) and the ones out of peak (1139 MHz and 1100 MHz, respectively – labelled as **out**, from out-of-peak). To this end, we slightly modified the substrate thickness in the simulations from 25 μm to 22 μm in such a way to have a simulated substrate resonance closer to the experimental one. The peak near 0 degrees in the previous version of the figure raised from numerical noise in the simulation; this has been avoided in the new version of the figure by improving the spatial resolution of the meshing grid employed in the FEM solver.

Changes in the draft: We modified **Figure 2** according to the reviewer request, indicating the frequencies we considered and producing new simulation results.

C2.5: The inset of Fig. 3 requires a scale bar. The experimental difference in Fig. 3(b-c) is only approximately 0.03 (rad), indicating low resolution. According to the authors' theory (Equation 1 in the supplementary material), the background signal's (longitudinal component) amplitude is an order of magnitude larger than the vortex signal (transverse component). How is the circulation path chosen, especially in Fig. 3(c) where there is a significant difference compared to simulations? Are there any solutions to reduce the background signal amplitude?

R2.5: We believe that a scale bar is already present in all the maps in Fig. 3 (now Fig. 4), both experimental and simulated ones, but we understand our inability to convey the reason for the different observed phase ranges in the experimental and simulated maps. Indeed, the phase range of old Fig. 3 (b-c) is very low, ranging 0.04 – 0.1 rad. As pointed out by the Reviewer, this is due to the presence of a strong background level. Moreover, the background amplitude depends on the driving frequency, being enhanced when resonant with the sharp substrate modes or by getting closer to the maximum of the BAWR generation band. Due to experimental constraints, we must acquire the interferometric signal close to or on top of the resonant peak position; while giving the best overall signal-to-noise ratio (and shorter acquisition time) this is also the spectral region mostly affected by the background, resulting in the low phase range observed in the experiment. We demonstrated (see Fig.2) that the phase range grows as soon as we measure out-of-resonance, although the lower signal leads to a very long acquisition time. The overall range of phases follows the general trend described above and, in fact, decreases while getting closer to the center of the BAWR generation band (~4 GHz - from Fig. 4(a) to (c) in our case); local and smaller variations are due to the specific frequency choice, given the possible superposition of strong, longitudinal modes with transversal ones, modifying the total background intensity. Having a thinner substrate and being less impacted by noise, the simulations are less affected by the background, which is only relevant at very specific frequencies separated by a wide free spectral range. This results in a larger phase range in the simulation data as reported in the maps in figure.

The circulation path has been chosen taking into account the geometrical center of the spiral-shaped top contact. In this task, we were aided by recording the laser reflectivity map, which

shows the central hole we made in the bottom contact, see Figure R2. We took care of possible shifts of the central singularity, which can slightly move from the center when a strong background signal is present (see Fig. S3 in the supplementary materials).

Figure R2: (a) Exploded view of the three layers composing the BAWR. (b): top view of the geometry of the BAWR. The interferometric map is usually acquired around the circular hole in the bottom contact (dashed square in (b)). The simultaneous acquisition of the acoustic field (see, for example the amplitude – (c)) and the reflectivity (d) allows one to determine the spiral geometrical center.

Finally, the background signal amplitude could indeed be reduced by considering a single-polished wafer substrate, as discussed in C1.5.

Changes in the draft: we added a discussion in the Vortex Characterization section (starting with “Operating between peaks...”) strongly underlining the reasons for the observed phase range and a possible way to decrease this effect. We further added new simulation results, collected in a new Figure (**Fig. S10**) in the Supplementary Note 7. See also our reply **R1.5**.

C2.6: In the supplementary material, does Fig. 2(a-b) correspond to cases where the propagation axes of the acoustic and optical waves are aligned and misaligned, respectively?

R2.6: Figure S2 and the relevant discussion in the text represent a pictorial way to understand the effect of an irrotational background signal within a vortex. The discussion therein considers the most general case and therefore applies to both acoustic or optical vortices. It is worth to note that we always assume an alignment of the propagation axes of the acoustic and optical waves: C1.3 depicts the standard interaction scheme we consider, where light impinges vertically upon the device top surface.

More specifically relevant to Fig. S2, one can imagine to represent a vortex as a point oscillating in a circular path in the complex plane: the topological charge shows as the number of complete revolutions made within one period of oscillation. An offset, constant in this trivial case, would offset the center of the circular path; then the number of revolutions within a period would be the same, but in terms of the global phase, they would show as a wiggling signal with a number of oscillations equal to the topological charge and with a reduced overall phase range.

While this is a trivial, illustrative idea, the results of a complete analytical model including an oscillating background are reported in panels Fig. S2(d-h), confirming this basic intuition. Moreover, a more realistic numerical model, approximating the system considering a Bessel’s function of the first kind with a helical phase (vortex) and an oscillating background is reported in Section B, further reproducing the same qualitative features.

Changes in the draft: we modified **Fig. S2** and the relevant description in the text to improve its clarity changing few sentences (from “within a single oscillation cycle...” on forth) in Supplementary Note 2A.

C2.7. In Equation 1 of the supplementary material, the background amplitude (B_{kg}) only significantly affects phase amplitude when it is around 1. When B_{kg} is much less than 1, it has minimal impact on the overall amplitude and phase; when B_{kg} is much greater than 1 (as in Question 5), the phase change becomes negligible. What criteria did the authors use to choose the value of B_{kg} , such as the 0-2 range selected in the supplementary material?

R2.7: The aim of subsection A and B in the vortex modelling section is to illustrate to the readers intuitive explanations for the reduced range and the phase oscillation we observed in the experiment. With this goal in mind we wanted to show regimes where the background is negligible ($B_{kg} \sim 0$) and stronger than the vortex amplitude ($B_{kg} \sim 2$); moreover, we wanted to show the general trend, which gives a reducing phase range for increasing B_{kg} .

Figure R3: Analytical amplitudes and phases for vortex with $\ell=1,2,3$ (column from left to right), with a varying B_{kg} from 0 to 10 in 0.5 steps.

As can be seen in Fig. R3, where we reproduced Figure S2 (c-h) of the supplementary information considering a B_{kg} from 0 to 10, the qualitative trend of both amplitude and phase is conserved, yet it is more difficult to appreciate the results due to the high density of closely spaced curves. We consider more illustrative to keep the original range in Fig. S2, although we added a sentence explaining the effect at very large backgrounds.

Changes in the draft: We added a sentence in Supplementary Note 2A starting with “We decided to consider...”

C2.8. On page 3 of the supplementary material: “In a first approximation, a circular BAWR can be approximated as as a drum resonator”—the phrase “as as” is redundant.

R2.8: We thank the reviewer for pointing out this typo that **we have corrected in the new version of the manuscript.**

C2.9. In Equation (2) of the supplementary material, there is no description of the constant C .

R2.9: We thank the reviewer for pointing out the typo in Equation (2) in the Supplementary Materials. **We have fixed it in the new version of the manuscript.**

C2.10. The supplementary material's Fig. 3 lacks figure numbers, although the caption includes (a) and (b), and no scale bar is present. The term “gs” is not explained. Fig. 3 shows that as B increases from 0 to 1, the phase singularity shifts from the center. For $l=2$, it appears that a single OAM-carrying ring beam ($B=0$) splits into two $l=1$ beams ($B=1$).

R2.10: We thank the reviewer for pointing out several issues within Figure S3 and we apologize for the errors in its caption, which was erroneously a duplicate of the one in the next figure.

Changes in the draft: the caption has been rewritten and Figure S3 improved, adding colorbar scales and choosing the background values in such a way to include very strong background signals (up to $B = 10$), as they can be interesting to observe. Furthermore, we expanded the discussion including some specific comments about vortex fields at large background levels (starting with “In particular, the increasing background level...”) in the Supplementary Note 2B.

C2.11. Although the total beam angular momentum remains 2, can it still be used for multiplexing in communication? I am curious about the effects when $B=2.5, 5, \text{ or } 10$.

R2.11: Regarding the specific observation of the Reviewer, we have elaborated this concept, producing a **new figure in the main text and two new figures in the supplementary material** and discussing the angular momentum-carrying optical beams and their orthogonality properties. Leaving the extended discussion to the manuscript, here we summarize our main conclusions:

- Optical modes generated exploiting mechanical-modulated differential reflection of a planar wavefront (see Fig. 1 in this document) using vortices with different topological charges are orthogonal to each other. The “orthogonality” of their scalar product matrix (Gram matrix - G_{ij}) depends on the ratio of the maximum of the real part of the vertical displacement ($\Delta z_M = \max [\Re(\Delta z)]$) versus the optical wavelength (λ_o). The particular shape of the matrix allows to assess its orthogonality calculating its determinant; as can be seen in Fig. 6 (a), reducing the displacement, decrease the orthogonality optical matrix while keeping essentially perfectly orthogonal acoustic vortex modes. The Gram matrix of the optical modes, explicitly reported for illustrative points, shows that a good degree of orthogonality is maintained even for displacements of the order of one tenth of the optical wavelength.
- The background displacement can contribute to a weakly decrease of acoustic modes orthogonality. As reported in Fig. S5 (b) (included in the new version of the supplement and reproduced below) a very strong background amplitude, twenty times larger than the vortex one ($B=20$) leads to small perturbations of the Gram matrix, with a corresponding determinant larger than 0.8.

Figure S5: (a): Determinant of the mechanical and optical Gram matrices as a function of the average maximum displacement of the acoustic vortices relative to the optical wavelength, $\Delta z_M/\lambda_0$. (b): Determinant of the mechanical Gram matrix as a function of the acoustic background displacement B . The complete Gram matrices are explicitly depicted for selected points (black dots).

Summarizing, our new results suggest that we have a large region of the parameter space where we can exploit the orthogonality of both mechanical and/or optical modes. Whereas a higher purity of the modes would be required, we can change our design as described in the previous answer to Reviewer #1 (see C1.4 and C1.5), for example by using a single polished substrate or implementing finger-like spiral contacts.

We agree with the reviewer that our manuscript draft should have delved more in the description of orthogonality properties of both mechanical and optical modes and we have therefore added significant material to the reviewed version of the manuscript.

Changes in the draft: we added one figure to the main text (**Fig. 3**), two figures in the supplementary information (**Fig. S4**, **Fig. S5**), added **more panel to figure S3** including the acoustic fields for larger background signals. The new figures have been described and discussed in a dedicated paragraph.

C2.11. The authors should provide simulation data for $l=6$ and $l=9$ in the supplementary material.

Changes in the draft: We ran the relevant simulations and **added their results** in the supplementary material (**Fig. S8**) with a dedicated description section (**Supplementary Note 6**).

C2.12. The authors may discuss and include more potential applications of such ultra-degree-of-freedom modulated vortex structured light such as ultra-capacity communications.

R2.12: We followed the Reviewer’s suggestion and **we included the reference:** “Z. Wan et al. Ultra-Degree-of-Freedom Structured Light for Ultracapacity Information Carriers, ACS photonics 10, 2149 (2023)” in the main text of our manuscript. We also added some discussion and some references reported in the aforementioned article in the Introduction (starting with “..., enriching the schemes...”).

3. Reviewer #3 (Remarks to the Author):

The authors have experimentally demonstrated the possibility to generate acoustic vortices with tunable topological charges up to 9 in a wide frequency band around 4 GHz on the basis of a single-contact piezoelectric bulk acoustic wave resonator. The resonator is composed of a ZnO

piezoelectric layer sandwiched between metallic contacts and it excites acoustic vortices in a sapphire substrate. The acoustic topologically charged waves are generated due to the fact that the resonator contact is modified into the shape of an Archimedean Spiral, which provides the required phase increment along a closed path around the center of the field. The vortex topological charge is then tuned by changing the driving frequency of the applied electric field. Besides, the generation of different acoustic topological charges was realized by making use of Archimedean spirals with multiple spokes at the fixed driving frequency. The experimental results were supported by numerical simulations with the use of commercial finite-element method solver (Comsol Multiphysics).

The authors claim that the presented device offers a new class of acousto-optic devices for generating optical vortices in the reflected beam from a normally incident Gaussian laser beam via acousto-optic interaction at the top surface. At that, they have not provided any measured data of the conversion efficiency of the incident optical beam into the produced optical vortex as well as of the generated optical intensity and phase distribution, and the corresponding topological charge. Also, nothing is said on the parameters of optical beams (beam waist, state of polarization, paraxial/non-paraxial). Even more importantly, the very description of acousto-optic interaction between acoustic and optical beams in the proposed scheme is also not presented in the manuscript. It is a critical drawback of the work because the physical process of the acousto-optic induced reflection accompanied by the formation of a vortex structure and transfer of OAM from longitudinally vibrating interface to the incident photons is far from trivial. In this connection, a number of fundamental questions arise:

C3.1. What is the regime of the acousto-optic diffraction (Bragg or Raman-Nath)?

How do the kinematic and dynamic conditions look like for an efficient coupling of the incident and diffracted beams?

C3.2. What is the dependence of the intensity of diffracted vortex beams on the waist of the incident beam, on the ratio of optical and acoustic wavelengths, on the state of incident polarization, and on the acoustic topological charge?

C3.3. Since the efficiency of the acousto-optic diffraction critically depends on the interaction length and the sapphire substrate has a rather small thickness of 425 μm , it is reasonable to assume that the energy efficiency of the device should be rather low that undermines the practical value of the proposed scheme, especially allowing for a number of the previously demonstrated highly efficient methods of optical vortex producing.

We thank the reviewer for pointing out a critical issue of our manuscript draft, namely the lack of discussion about the generation of optical beams carrying optical angular momentum. We appreciate and duly note that none of the reviewers contested our observation of acoustic vortices, whose investigation covered a substantial portion of our study and whose realization at super-high-frequency unveils novel and interesting applications for acoustic and hybrid technologies.

R3.1: The coupling scheme is **explained in the answers to Reviewer #1 (Sec. R1.3)**; it is based on the phase shift of the reflected light due to the varying optical path created by the out-of-plane mechanical displacement or photo-elastic effects. The acousto-optical interaction happens at the device top surface. Since the light and acoustic waves propagate along the same axis with a very short interaction length, we are, therefore, strictly neither in the Bragg nor in the Raman-Nath conventional diffraction regime (and we did not make any observation about kinematic and

dynamics conditions). Although we use a different coupling scheme, in principle our device could also work for co-propagating acoustic and optical waves within the substrate, which would enable a diffraction scheme in the Bragg regime. Discussing it is out of the scope of this paper but it could be relevant for future investigations.

R3.2: Generally speaking, we observed that the intensity of the reflected light depends on the material reflectivity (which is favoured by employing metallic contacts as in our case) and, for angular momentum transfer, the ideal optical beam waist would be as large as the acoustic vortex lateral extension (roughly the dimension of the central hole in our design, at least for the lowest topological charges).

R3.3: As stated by the Reviewer, even if our coupling mechanism is different than the conventional diffraction scheme, it is essential to define the interaction efficiency. In our case the generation of orthogonal OAM optical beams depends on the ratio of the optical wavelength (λ_0) and out-of-plane displacement (Δz), creating orthogonal or quasi-orthogonal modes for different acoustic vortice topological charges as can be seen in Fig. 5 in this letter and **in the answers to Reviewer #2**.

The acousto-optic coupling is indeed proven by our interferometric result, but we recognize that it needs to be complemented with a more detailed description of the optical fields to illustrate their properties as angular momentum carrying optical beams.

Changes in the draft: We added a new figure to the main text (Fig. 3) with an appropriate discussion greatly extending the Device Concept section (starting with “The directly generated...”). We added two new figures in the supplementary material (fig. S4 and S5), discussed in a dedicated section Supplementary Note 2C. The additional material complements the treatment of acoustic vortices, describing the optical fields generated by illumination of the acoustic vortices with planar wavefronts, including their orthogonality properties and their robustness within the device parameter space.

C3.4. Besides, some of these approaches also allow for a dynamic electronic control of the generated OAM, including those based on liquid-crystal spatial light modulators [A. Forbes et al., "Creation and detection of optical modes with spatial light modulators," Adv. Opt. Photon. 8, 200-227 (2016)] or acousto-optic interaction in optical fibers [M. Yavorsky et al., "Photon-phonon spin-orbit interaction in optical fibers," Optica 8, 638-641 (2021); Jiafeng L. et al., "Recent progress of dynamic mode manipulation via acousto-optic interactions in few-mode fiber lasers: mechanism, device and applications", Nanophotonics, vol. 10, 2021, pp. 983-1010].

R3.4: We thank the Reviewer for pointing out extremely interesting literature which **we added to the new version of the manuscript**. Nevertheless, we strongly believe that the acousto-optic device we present here strongly differs and offers new functionalities with respect to the literature here included and referenced in the mentioned manuscripts.

First of all, most of the works exploiting acousto-optic interaction in optical fibers investigate vortices with low topological charge (± 1 in the theoretical work of [Yavorsky et al.] and in in most of the works cited in [Jiafeng et al.]), suggesting a very different regime of operation with respect of the device introduced in our manuscript.

Furthermore, and even more relevant, our device is the first acousto-optical one for OAM transfer operating at extremely high acoustic frequency, exceeding 1 GHz. This is more than one order of magnitude higher than the best purely acoustic vortex and more than two orders of magnitude

higher than acousto-optic devices implemented within an optical fiber, see **RF frequency** column in Table 1, extracted from [Jiafen et al.].

Acoustic mode	Fiber type	Incorporations	RF frequency	Resonant wavelength	AO length	Mode conversion	Demonstration	Refs.
Flexural mode	TMF	-	8 MHz	488 nm	7.5 cm	LP ₁₁ mode	Frequency shifter	[43]
	TMF	Two AOMCs	2.51 MHz	1550 nm	4.5 cm	LP ₁₁ mode	Comb filter	[55]
	SMF	Taper	1-1.5 MHz	1532-1550 nm	10 cm	Cladding modes	Attenuation filter	[56]
	SMF	Cuneal transducer	1.95-2.45 MHz	1550 nm	8.3 cm	Cladding modes	Tunable notch filter	[59]
	SMF	Parallel AOMCs	0.9001-1.066 MHz	1490-1610 nm	10 cm	Cladding modes	Broadband coupler	[61]
	SMF	SNS structure	2.384 MHz	1527.7 nm	70 cm	Cladding modes	Band-pass filter	[63]
	SMF	FBG	1.3 MHz	1541.5 nm	1.7 cm	Reflecting modes	Reflection switch	[68]
	SMF	FBG	1.3 MHz	1541.5 nm	1.7 cm	Cladding modes	FBG phase-matching	[69]
	SMF	LPG	2.43 MHz	1543.3 nm	20 cm	Cladding modes	Vibration measurement	[71]
	SMF	Taper	0.9 MHz	1550 nm	4 cm	Cladding modes	Vibration measurement	[72]
	PCF	-	7.4 MHz	633 nm	48 cm	LP ₁₁ mode	Birefringence analysis	[76]
	SMF	MZI	0.91-0.99 MHz	1561.6-1568.9 nm	6 cm	Cladding modes	Tunable laser	[79]
	SMF	CMB	0.575-2.934 MHz	1641.4-2127.6 nm	24 + 34.3 cm	Cladding modes	Tunable laser	[82]
	EDF	-	1.2 MHz	1550 nm	8 cm	Cladding modes	Q-switch laser	[83]
	SMF	Taper	1.23 MHz	1529.5-1532.5 nm	23.7 cm	Cladding modes	Mode-locked laser	[90]
	SMF	Polarizer	1.039-1.069 MHz	1539-1571 nm	13.5 cm	Cladding modes	Polarization conversion	[92]
	TMDCF	Orthogonal PZTs	600-900 kHz	1520-1570 nm	30 cm	OAMs	OAM generation	[93]
	FMF	Orthogonal PZTs	0.8227-0.8289 MHz	633 nm	4 cm	OAMs	OAM generation	[96]
	TMF	-	2.687/2.749 MHz	1040.8 nm	50 cm	CVBs	CVB generation	[97]
	E-FMF	FSK	738/753 kHz	1550 nm	25 cm	OAM	OAM switching	[105]
E-FMF	FSK	467.21/485.83/752.6/777.3 kHz	1064 nm	14.6/15 cm	LP ₁₁ /LP ₂₁ modes	HOM generation	[107]	
FMF	Taper	824/841 kHz	1572 nm	10 cm	LP ₂₁ mode	Mode-locked HOM laser	[108]	
DCF	-	670 kHz	1070 nm	-	LP ₁₁ mode	High power HOM laser	[109]	
E-FMF	FSK	726/742 kHz	1532.9/1534.3 nm	25 cm	OAM	OAM switching dynamics	[112]	
Longitudinal mode	SMF	FBG	8.02 MHz	1526.5 nm	0.3 cm	Reflecting modes	Super-lattice modulation	[65]
	SMF	FBG	10/10.7 MHz	1528 nm	0.3 / 1.2 cm	Reflecting modes	Tunable reflector	[66]
	SMF	FBG	2/12.4/5/6/10 MHz	1550 nm	1 cm	Reflecting modes	Super-lattice modulation	[67]
	SMF	FBG	1/2.66/5.5 MHz	1543.2 nm	5 cm	Reflecting modes	Q-switch laser	[84]
Torsion mode	HBSMF	Two shear PZTs	1.337 MHz	1550 nm	60 cm	Polarization modes	Polarization filter	[113]
	E-TMF	Twisting	3.24/3.4 MHz	1300 nm	25.5 cm	LP ₁₁ mode	Twist effect analysis	[114]
	HBSMF	PBS	2.638-2.788 MHz	1530-1620 nm	82 cm	Polarization modes	Tunable filter	[115]
HBSMF	-	1.189 MHz	1320 nm	49.8 cm	Polarization modes	Band rejection filter	[117]	

Table 1: Summary of the devices based on acousto-optic interactions (AOIs) in fibers with different vibration modes. From [Jiafeng L. et al., "Recent progress of dynamic mode manipulation via acousto-optic interactions in few-mode fiber lasers: mechanism, device and applications", Nanophotonics, vol. 10, 2021, pp. 983-1010]

The other interesting review pointed out by the Reviewer [Forbes et al.] illustrates devices for the reconfigurable generation of optical vortices with various and large topological charges using liquid crystals. Even if dynamical operation frequencies are not directly mentioned in the review, it is a widely assessed fact that liquid crystal devices can operate at most at 10s of kHz frequency (state-of-the-art devices) with routine operation at 1 kHz or less. Again, we respectfully think that this range or operation strongly differs from the one introduced here, being the operating frequencies in liquid-crystal devices widely surpassed by our BAW, operating at frequencies in the GHz range.

The wide topological charge tuning, the highest RF frequency ever reported, and the unexplored generation mechanism make us believe the importance of introducing our acousto-optic device, which we think is an interesting addition to the wide ensemble of devices capable of generating and modulating orbital angular momenta.

Change in the draft: We highlighted our findings stressing the main differences with other approaches in the literature, by adding an extended paragraph in the Introduction, starting with "with distinct characteristics...". Moreover, we improved our literature review, adding more references in dedicated sentences in the Introduction, starting with "..., enriching the scheme...".

Q3.5 As a second critical comment, it is stated in the text that the proposed approach allows one to create optical vortices with time-varying orbital angular momentum. I believe this is an erroneous conclusion drawn from a misunderstanding of the essence of the concept of "time-varying optical vortices (as well as spatio-temporal vortex beams)". In fact, these are the light

pulses (polychromatic!) with the well-defined OAM that is changing in the space-time domain due to the inherent complex evolution of spectral components of the wave-packet [L. Rego et al., “Generation of extreme-ultraviolet beams with time-varying orbital angular momentum, Science, 364, (2019); Konstantin Y. Bliokh, “Spatiotemporal Vortex Pulses: Angular Momenta and Spin-Orbit Interaction”, Phys. Rev. Lett. 126, 243601 (2021)]. Instead, like any other acousto-optic device, the proposed in the manuscript scheme simply enables switching between generated monochromatic optical vortices with different values of their fixed OAM by means of varying the applied driving frequency.

R3.5: We thank the Reviewer for this precious observation, which made us reflect on the definition we have used to describe our OAM carrying beams. Indeed, we agree that our definition was misleading; we, therefore, changed it to something more pertinent and routinely used by the scientific community operating in the field. In our structures, the incoming optical field is oscillating at its own photon frequency (i.e. ~563 THz for the green laser employed) and simultaneously rotating at the mechanical frequency (0.5 – 7 GHz, depending on the rf drive) – hence, the energy shifts of the reflected beam are very small. The use of the term “time-varying” may be misleading, since it is conventionally referring to a different phenomenon. To improve the clarity and correctness of our manuscript, we have then illustrated the formation of OAM carrying optical beams as resulting from an effective “mechanical” spiral phase plate, operating in reflection and dynamically rotating at the excited mechanical frequency, combining different results reported in literature for macroscopic optical elements [2-3] and implementing them in a new mesoscopic and on-chip system. The phase plate elevation and, therefore, its effectiveness for modulating the reflected beam depends on the ratio between the mechanical displacement Δz and the optical wavelength λ_0 ; as in the answer to Reviewer #2 (R2.11), this produces orthogonal optical beams for a wide parameter range, both considering the acoustic vortex and background signal amplitudes.

This description of our system is advantageous for a clear understanding of the physical mechanism underneath and facilitates easy comparison with and validation by experiments related to angular Doppler shift involving rotating macroscopic optical elements.

As an example, the illumination with a wide, homogeneous laser beam upon a rotating optical element with a weakly embossed pattern (with a symmetry with $\ell = \pm 18$) shows the same ℓ angular momentum embedded in the reflected beam, as obtained by measuring the rotational Doppler effect [4].

To clarify these points to the reader, we introduced the following text to the introduction:

The vertical profile of the acoustic field is qualitatively similar to a reflective spiral phase plate [29], rotating upon its axis at the mechanical frequency; its interaction with an illuminating wavefront reflects a wave with a RF rotational angular momentum [30], realizing a system similar to the ones employing macroscopic spinning optical elements for the detection of rotational Doppler’s effect [31], albeit at super-high rotational frequency.

C3.6: To summarize, the used in the paper method of generating acoustic vortices has been previously suggested and already experimentally realized, the acousto-optically-induced generation of optical vortices is totally not described, and creation of the time-varying optical vortices seems to be miss-attributed. From these perspectives, for the present paper, there is no novel finding and enough impact for publication in Nature Communication.

R3.6: We respectfully disagree with the Reviewer's opinion even if we understand the motivation behind the Reviewer report, who identified weak points in the previous version of our manuscript. A general reply to C3.6 is representative of the whole revision work we performed, applying also to the issues raised by Reviewer #1 and #2, who showed a more positive judgment of our work, although highlighting its limitations as described in the previous version of the manuscript. We believe we have now successfully addressed each and all of their criticism and observations in the new version of the manuscript, greatly improving the presentation of our report.

Noting again that none of the Reviewers mentioned any possible objection regarding the generation of super-high-frequency acoustic vortices on a solid-state platform, which we consider one of the main achievements of our investigation, we clarified the following points:

- We described the generation mechanism of acousto-optic interaction, showing that is more akin to a dynamically rotating phase plate than to acousto-optic diffraction (as in optical fibers), making it novel and interesting for the scientific community.
- We strongly re-state that, to the best of our knowledge, no acousto or acousto-optic device for AOM transfer has so far been reported for operation at GHz frequencies, making our system novel and interesting for various application, also considering the large topological charge tuning, which is not a common feature in most of OAM-based devices.
- We are grateful for pointing out possible flaws in our definition of time-varying OAM beams. Understanding its use as a possible misleading term, we reworked our definitions, expanded the mode description and include relevant literature to help the reader understanding the working principle of our system.
- Finally, we addressed the possible limitations of our device (including, for instance, the low ratio $\Delta z_M/\lambda$ achieved in the present configuration), discussing some strategies to overcome them, opening the route to future research avenues.

Changes in the draft: we modified and expanded the Introduction, citing manuscripts which are relevant to the discussion. We added a new figure in the main text (**Fig. 3**), two new figures (**Fig. S4 and S5**) in the supplementary information and a relevant discussion describing the optical beams that can be generated by the acoustic vortices, considering their properties and limitations.

We hope that the major revisions we performed, aligning our manuscript to the Reviewers' requests, will make it suitable for publication in a prestigious journal such as Nature Communications

References

- [1] H. Tian, J. Liu, B. Dong, J. C. Skehan, M. Zervas, T. J. Kippenberg & S. A. Bhave, Hybrid integrated photonics using bulk acoustic resonators, Nat. comm. 11, 3073 (2020)
- [2] Y. S. Rumala, Optical vortex rotation and propagation from a spiral phase plate resonator with surface reflective coating, Opt. Lett. 45, 1555 (2020)
- [3] C. Zhang & L. Ma, Millimetre Wave with Rotational Orbital Angular Momentum, Sci. Rep. 6, 31921 (2016)
- [4] M. J. P. Lavery, F. C. Speirits, S. M. Barnett, & M. J. Padgett, Detection of a spinning object using light's orbital angular momentum Science 341, 537 (2013)

June 9, 2025

Answer to the referee's questions

Dear Editor, dear Reviewers,

We thank the Reviewers for reporting their recommendations on our revised manuscript. We appreciate that both Reviewer #2 and Reviewer #3 recognize the impact of our results as the first demonstration of on-chip tunable acoustic vortices at GHz frequencies, which are suitable for light interaction and generation of optical beams carrying orbital angular momentum and especially, citing Reviewer #3, “at previously non-reported GHz frequencies”.

We also thank Reviewer #1 for her/his careful consideration of the revised version of our manuscript. He/she mentions that “one key novelty of this work, in my view, lies in achieving this effect [i.e. GHz AOM generation] dynamically or through electrical tunability. Accomplishing this in a micrometer-scale structure is far from trivial and represents a significant challenge.” At the same time, the Reviewer asks for further clarifications. We address below her/his concerns, hoping to further improve the clarity of our message and highlighting its most important features to have an even better version of our manuscript. The modifications introduced in the main text are highlighted in color in an attached pdf document.

Reviewer #1 (Remarks to the Author):

Thanks for the authors' clarifications and efforts for answering my concerns.

We also thank the Reviewer for having given us the opportunity to further expand and improve the new version of the manuscript.

R1. It seems that the authors adopt a broader definition of acoustic-optic interaction, whereas I prefer to view it as a type of boundary deformation effect. Under the authors' definition, nearly all natural materials could exhibit “acoustic-optic interaction”. Since in general, optical waves reflect at boundaries or interfaces, and any boundary deformation influences the reflection phase. It would be reasonable if the authors explicitly stated that they are employing a generalized acoustic-optic interaction before using the term. However, I still recommend that they consider an alternative term to describe the mechanism in their paper to avoid potential confusion for readers.

We agree with the Reviewer that we employ a broader definition of acousto-optic interaction. This was originally done to comprise all possible interactions of light and vibrations in our device, including the “moving boundary”, which is the dominating one in our structures, as well as the “photo-elastic” interaction between waves propagating within the substrate, which is not being considered in this experiment. For the sake of clarity, we have clarified our definition in the revised version of the manuscript.

Changes in the manuscript:

We defined our interaction as “generalized acousto-optic interaction” specifying that the coupling mechanism in our experiment is essentially the moving boundary effect. Modifications have been performed in the **Introduction** (“OAM acousto-optics” → “generalized OAM acousto-optics interaction”, “its interaction” → “its generalized interaction” and “, more specifically based

*on a moving-boundary effect”) and in **Device Concept** (“the main acousto-optics coupling, ...”
→ “the main acousto-optics coupling, ... due to the moving boundary effect”).*

R2. The primary concern remains whether a vortex of surface out-of-plane vibrations can truly induce a vortex (or orbital angular momentum, OAM) in a reflected optical beam. First, note that generating a reflected optical beam with OAM is relatively straightforward. For example, by fabricating spiral structures (e.g., q-plates) on a metal surface, one can impose the desired phase on the incident beam, thereby producing a reflected OAM beam. Similarly, exciting a surface vortex is not particularly challenging, as previous studies have demonstrated that spiral nanostructures can generate plasmonic vortices carrying OAM — a concept that should also apply to elastic waves. One key novelty of this work, in my view, lies in achieving this effect dynamically or through electrical tunability. Accomplishing this in a micrometer-scale structure is far from trivial and represents a significant challenge.

We agree with the Reviewer that generating a light vortex with a static “phase structured” reflector, being a spiral-phase plate, a metasurface or a spatial light modulator can be a straightforward concept; moreover, we acknowledge that other systems have been reported, where spiral structures lead to the generation of vortex excitations. However, we must add that applying this concept to elastic waves is already far from trivial, given the specificity of every physical system. On top of that, our system leverages on the dynamic dimension by rotating acoustic fields at **GHz frequencies**, far surpassing other reports of acoustic vortex in the literature. Furthermore, its properties, such as the topological charge, are tunable and electrically addressable.

We also agree that devising our device was indeed a significant challenge but we must add that also gives significantly impacting results, acting indeed as a very peculiar spiral phase plate which (i) is defined on a chip, (ii) rotates upon itself at GHz frequency and (iii) with an overall phase shift that can be externally controlled by changing the drive conditions. We think these properties make our device unique and interesting for a wide research community.

R3. However, as mentioned in my previous response, for this primary concern, the authors have not yet provided any direct experimental observation of an optical vortex (or OAM). Instead, they measure surface vibrations by focusing a beam on each position of the metal plate and infer the presence of OAM in the reflected optical beam using the Huygens–Fresnel principle, as shown in Fig. 3. Considering this as an experimental study, I remain skeptical about whether it can truly generate OAM in an optical beam in practice.

R3a. Firstly, the derivation based on indirect experimental measurements looks not entirely solid.

Here, we first note that while we report the formation of acoustic vortices, we do not claim the generation of optical vortices but rather the transfer of OAM from the acoustic vortex to an optical beam (which is, e.g., sufficient for several technological applications). Furthermore, the interferometric experiments are, in fact, a **direct** measurement of the optical phase across the cross-section of the reflected beam, as will be further elaborated in point R4 below.

R3b. Regarding the “phase reduction” effect (see R1.5), I sincerely appreciate the authors’ response that the substrate significantly alters the phase distribution, leading to substantial discrepancies between simulation and experimental data. However, the authors acknowledge that no improvements can fully eliminate this background effect. This raises a critical issue: while simulations suggest the possibility of generating an optical

vortex, the experimental data deviate significantly from these predictions. How can we confidently conclude the presence of an optical vortex based solely on simulations? Moreover, the measured reflected phase shows only minimal variation — e.g., 0.03 rad in Fig. 4(b) and 0.02 rad in Fig. 4(c). Such a small phase difference seems insufficient to induce a clearly defined optical OAM beam.

We understand that one of the subtleties of our report is the “phase reduction” effect, which has been one of the main revised sections of the first version of our manuscript. As a first point, we probably failed to explain the differences between the experiments and simulations in Figs. 1 and 2 and the ones in Fig.4. Limitations in the available computational memory do not allow us to fully reproduce the experimental background effect for all experimental data (i.e. we cannot simulate the whole sapphire thickness). However, we believe that the simulations well reproduce the experimental features without discrepancies, as elaborated in the discussion of the acoustic background effects in Sec. “Device Concept”.

To further clarify this point, Fig. R1 redraws, for convenience, the main results of Fig. 1 and Fig. 2 (the latter redacted for a small mistake in the phase units and in one of the simulation datasets), for excitation on a strong resonance (“on-peak”). As described in the text, the phase excursion becomes strongly reduced under this excitation condition. Plotting the two data sets together, one can clearly appreciate the good agreement between experimental and simulated data, thus validating our computational platform.

Figure 2 of the main text also reports experimental and calculated phase profiles recorded “off-resonance”, i.e., at a frequency shifted with respect to the substrate longitudinal resonances. Both show an increased phase excursion as compared with the “on-resonance” case, with the simulations almost fully recovering the whole $[-\pi, \pi]$ range expected for a $\ell = 1$ vortex.

Unfortunately, the experimental signal amplitude in the “off-resonance” measurements, as well as in the simulation, are much smaller than for on-resonance excitation, thus probing them would make the experiment very time-consuming due to the reduced signal-to-noise ratio (SNR). For this reason, the measurements and simulations under “off-resonance” conditions were only performed to understand the role of the substrate in connection to Figs. 1 and 2 of the main text, where we established the differences between the acoustic phase profiles in spectral regions with a strong or weak background. The role of a varying background has also been considered in Suppl. Mat. Sections II.A and II.B, where we showed analytical and numerical models for the acoustic fields without and with different background amplitudes, as well as in Suppl. Mat. Section II.C, where we report on the acoustic mode orthogonality with different backgrounds, providing a comprehensive overview of the effect.

Figure R1: Comparison between experimental (upper left panel) and simulated (lower left panel) acoustic fields for a vortex with topological charge equal to 3. The right panel compares the experimental and calculated angular phase profiles extracted from the field maps.

Furthermore, we believe that the background, despite masking some of the acoustic features, does not have a strong effect on the optical mode orthogonality. This is indeed a point that we clarify here and in the revised text, taking the opportunity offered by the Reviewer.

Figure R2: Sketches for representing the reflected wavefronts upon illumination of a perfectly reflected surface with a plane wave beam. General reflection (a), considering a constant, oscillating background (b) or an oscillating displacement pattern (c).

Going back to the general concept of a wavefront reflected by a surface, we can describe the reflected field E_r in polar coordinates (cf. Fig. R2a) as:

$$E_r(r, \theta) = \mathcal{R} \cdot E_i(r, \theta) e^{2\pi i \frac{2\Re[\Delta z(r, \theta)]}{\lambda_o}}$$

where E_i is the incident field, \mathcal{R} the surface reflectivity, λ_o the optical wavelength and Δz the vertical component of the acoustic field. Assuming a perfectly reflecting surface (i.e. a metallic surface under the plasma frequency, $\mathcal{R} = 1$) and a planar incident wavefront ($E_i(r, \theta) = E_i$), the spatial dependence of the reflected field depends only on the phase term $\Re[\Delta z(r, \theta)]$.

If we assume an oscillating constant background, $\Re[\Delta z(r, \theta)] = \Delta Z_B$, as depicted in Fig. R2b, an incident planar wavefront will result in a reflected planar wavefront. Conversely, a complex background $\Re[\Delta z(r, \theta)] = \Delta Z_V(r, \theta)$ imparts a spatial phase profile to reflected light beam and, as discussed in the main text, transfers OAM to it. The generalization for a superposition of oscillating background and vortex pattern is straightforward, the former yielding a multiplicative constant in the expression for reflected field:

$$E_r(r, \theta) = E_i e^{2\pi i \frac{2\Delta Z_B}{\lambda_o}} \cdot e^{2\pi i \frac{2\Delta Z_V(r, \theta)}{\lambda_o}}$$

In Sec. 2.C of the Supplementary Information we further generalize this formulation considering the oscillating background as mostly constituted by $\ell = 0$ modes, which can, in essence, be described by a with zero-order, radial Bessel's function of the first kind. Therefore, while the $\ell = 0$ background acts on the radial profile, it does not impact the θ spatial profile, thus preserving the properties of angular momentum.

Figure R3: Gram matrix of reflected optical fields calculated considering (a) a purely vortex displacement pattern and (b) a superposition of vortex and a background with the same amplitude (b)

To further convince ourselves that the background does not influence the OAM transfer and the orthogonality properties of the reflected beams, we have calculated the Gram matrices in different conditions, as depicted in Fig. R3. At first, we only considered light reflected from an oscillating background, described as a zero-order, radial Bessel's function of the first kind. The reflected fields do not depend on ℓ and, therefore, they cannot form a base in the angular momentum parameter (not shown). When considering only vortices, we find the Gram matrix of Fig. R3 (a). Here, we have considered a displacement amplitude equal to half an optical wavelength. Finally, when we consider a superposition of vortices and a background with the same amplitude, we find the Gram matrix of Fig. R3 (b). As can be seen, the orthogonality properties are well preserved, validating the analytical argument we described above.

R3b(ii). Reduction/elimination of background effects

We had already discussed in the previous version of the manuscript strategies to reduce the background contribution, as stated in the passage on page 5:

“Broadband operation is granted by the specific geometry of our device with a solid contact whose outermost rim is carved like a spiral. A properly shaped BAWR consisting of multiple metallic fingers arranged in a spiral guise would lead to sharper resonances and different regime of operations. The spiral arrangement would, in fact, impose stricter in-plane resonant conditions, with the finger spacing determining the acoustic wave generation efficiency and leading to higher purity vortices while sacrificing the frequency tunability of the topological charge.”

To further substantiate these claims, we also introduced after the first round of revision additional FEM simulations in the Supplementary Information (cf. Fig. S10 and reproduced in Fig. R4) The results support this hypothesis by showing phase profiles of the very same device operated at the same frequency yet with perfectly reflecting (Fig. R4a) or perfectly absorbing substrate backside (Fig. R4b).

In the previous revision, we further complemented these results by illustrating simulations similar to the ones of Fig. R5 in the response letter. They show profiles for an $\ell = 1$ acoustic vortex obtained using an Archimedean spiral finger contact and assuming a non-reflecting back facet of the sapphire substrate. The size of the simulation cell is the same as the ones reported in the main text, the acoustic frequency is 500 MHz, and we again neglected the in-plane anisotropy of the substrate. As expected, by strongly suppressing the generation of $\ell = 0$ mode, all the features of a standard vortex are recovered yielding the expected, monotonically increasing phase from $-\pi$ to π radians (as already obtained in Fig. S10), the central singularity and the ring-shaped amplitude.

We believe that both strategies (i.e. non-reflecting substrate back surface and spiral-shaped top) enable the generation of vortex patterns with a higher OAM purity, while reducing the acoustic amplitudes and broadband AOM tunability in comparison to the Archimedean spirals investigated in the main text.

Changes in the manuscript:

1. We added a sentence better describing the limit of the simulations in completely reproducing the substrate effect on the background generation, page 4: “*The lack of quantitative ...*”
2. We described the effect of the background displacement on the optical mode orthogonality by adding a paragraph in the Supplementary Information, **Supplementary Note 2.C**. Moreover, we added a sentence at the end of the Device Concept section: “*the latter ...*”

3. Along with the previously included results S10. S10 we added the numerical simulations of Fig. R5 extending the section in the Supplementary Information, **Supplementary Note 7** with a dedicated new figure, Fig. S11. Moreover, we added a sentence (“*Even under the strong ...*”) and a paragraph at the end of the Vortex Characterization section, starting with “*The phase excursion...*”

R4. Secondly, the experimental setup in this work is essentially a home-made laser vibrometer designed for micrometer-scale structures. As we know, a laser vibrometer measures surface vibrations by detecting Doppler shifts in the reflected laser beam caused by surface displacement. Vortex patterns similar to those presented by the authors are often observed in surface vibration measurements, particularly in wave interference from multiple scatterings or metamaterials — though typically at the millimeter scale. However, while these vortex patterns are detected optically, they correspond to vortices in the elastic field rather than optical vortices. This distinction is crucial, as the experimental setup is mainly designed for spectrally and phase-resolved detection of surface displacement rather than the direct observation of optical OAM. Detecting OAM should be achieved through independent measurements, such as Fork grating interferometry .

To summarize, can the authors conduct an experiment that directly measures the optical vortex (or OAM)? If the authors can provide such results, I think it would effectively address technical concerns regarding this aspect of the study.

Again, as previously mentioned, we do not claim the formation of optical vortices but rather the transfer of OAM from an acoustic vortex to an optical beam. We agree with the Reviewer that we do not report a direct measurement of the OAM of an **extended light beam** carrying angular momentum.

We note, however, that, on the one hand, the employed vibrometer is in fact an optical Michelson interferometer: the primary measurement quantity is thus the phase of the reflected beam, from which one infers the amplitude of the acoustic displacement. On the other hand, we do believe that standard OAM interferometry can neither give additional information nor be more easily implemented and sensitive than our scheme, given the high-frequency of the generated OAM-carrying beam. For that purpose, we introduced a new section on the supplement (Supplementary Note 8) analyzing the prospects of fork interferometry on our structures. In fact, conventional Fork grating interferometry would require imaging of the interfered beam profile at the sub-nanosecond time scale synchronized with the acoustic field. This would require a dedicated setup at the forefront of the modern techniques for time-resolved light detection. One could then rely on an image reconstruction “pixel-by-pixel” in the reciprocal space as a viable alternative; but this is exactly what our interferometric technique does, although operating in real space and directly mapping the beam phase synchronized with the mechanical field. For completeness, we also present in Supplementary Note 8 an alternative stroboscopic approach for Fork interferometry based on pulsed illumination: the implementation and sensitivities, however, are not expected to be higher than for the Michelson approach used in the text.

Assuming that our phase maps directly represent the vertical acoustic fields oscillating at GHz frequency – which can be measured through an already quite elaborated characterization setup – then just using widely assessed hypothesis, such as the Huygens-Fresnel principle, one can obtain the phase profile of a planar wavefront reflected by the device. We agree that a GHz-

stroboscopic, direct image of the OAM-carrying light beam would represent an interesting result, but unfortunately it would come with challenging experimental requirements, which would lead to the same “image reconstructed” profile we are evaluating in our experiment.

Changes in the manuscript:

We expanded the discussion on interferometric characterization, explicitly citing Fork interferometry and introduced a new supplementary section (**Supplementary Note 8**), where we discuss a potential procedure for stroboscopic Fork interference and predictions of the corresponding patterns under dynamic acoustic vortices.

In conclusion, we are glad that the previous version of the manuscript already fully satisfied Reviewer #2 and #3. We appreciate that Reviewer #1 recognized some of the main results of our manuscript, namely the all-electrical control and topological charge tunability of the acoustic vortices, to which we add the highest ever reported frequency, surpassing 1 GHz. With the present modifications and the accompanying clarifications in this response letter, we believe that we further improved our manuscript by, (i) rephrasing the definition of acousto-optic interaction, (ii) demonstrating that the acoustic background, although masking the traditional vortex shape, does not impact on the orthogonality of OAM-carrying optical beams and (iii) clarifying the difference between our interferometric characterization techniques and other approaches based on imaging, such as Fork-grating interferometry. We hope that the with the modification introduced in the revised version of the manuscript will receive recommendation for publication also from Reviewer #1.

In the resubmission package, we include a pdf version of the manuscript highlighting the changes introduced to address the reviewers' comments.

With our best regards,

Alessandro Pitanti and Paulo V. Santos

July 1, 2025

Answer to the referee's questions

Dear Reviewer,

We are glad that the Reviewer recommends the publication of the manuscript after the changes introduced during the last review. We are also thankful to the reviewer for pointing out formatting issues (e.g., missing reference), which will be addressed below.

Reviewer #1 (Remarks to the Author):

I have no further technical comments and can recommend the paper for publication now.

But before final acceptance, please address the following formatting issues:

R1. The “Merged File containing manuscript text” does not incorporate all tracked changes (highlighted in red) from the “Article File with Track Changes”. There appear to be processing errors, as some missing changes are simply marked with a “1”. Please ensure the final text file is generated accurately with all revisions included.

2. On Page 7 of the “Article File with Track Changes”, the final paragraph (highlighted in red) before the conclusion section contains missing references, indicated by an “?” symbol (As mentioned above, this paragraph is not included in the “Merged File containing manuscript text”). Please verify and complete the references accordingly

We are glad that the manuscript now satisfactory addresses the concerns of referee #1, who now recommends publication. We apologize for the formatting mistakes, which led to the marks “1” as well as a missing citation in the main text. These have been corrected in a revised version uploaded to the journal's server.